# Effective Halogen-Free Flame-Retardant Additives for Crosslinked Rigid Polyisocyanurate Foams: Comparison of Chemical Structures

**DOI:** 10.3390/ma16010172

**Published:** 2022-12-24

**Authors:** Johannes U. Lenz, Doris Pospiech, Hartmut Komber, Andreas Korwitz, Oliver Kobsch, Maxime Paven, Rolf W. Albach, Martin Günther, Bernhard Schartel

**Affiliations:** 1Leibniz-Institut für Polymerforschung Dresden e.V., Hohe Str. 6, 01069 Dresden, Germany; 2Covestro Deutschland AG, Kaiser-Wilhelm-Allee 60, 51365 Leverkusen, Germany; 3Bundesanstalt für Materialforschung und-Prüfung (BAM), Unter den Eichen 87, 12205 Berlin, Germany

**Keywords:** flame retardant, dibenzo[d,f][1,3,2]dioxaphosphepine 6-oxide, BPPO, 9,10-dihydro-9-oxa-10-phosphaphenanthrene-10-oxide, DOPO, polyisocyanurate, PIR, rigid foam, cone calorimeter, Pudovik reaction

## Abstract

The impact of phosphorus-containing flame retardants (FR) on rigid polyisocyanurate (PIR) foams is studied by systematic variation of the chemical structure of the FR, including non-NCO-reactive and NCO-reactive dibenzo[d,f][1,3,2]dioxaphosphepine 6-oxide (BPPO)- and 9,10-dihydro-9-oxa-10-phosphaphenanthrene-10-oxide (DOPO)-containing compounds, among them a number of compounds not reported so far. These PIR foams are compared with PIR foams without FR and with standard FRs with respect to foam properties, thermal decomposition, and fire behavior. Although BPPO and DOPO differ by just one oxygen atom, the impact on the FR properties is very significant: when the FR is a filler or a dangling (dead) end in the PIR polymer network, DOPO is more effective than BPPO. When the FR is a subunit of a diol and it is fully incorporated in the PIR network, BPPO delivers superior results.

## 1. Introduction

Lightweight foams are of growing importance in materials design due to the fact that mass, energy consumption, and CO_2_ emissions correlate. Weight-saver strategies are abundant in mobility applications and lead to significant energy savings. These strategies involve the use of foams made from metals [1], thermoplastics [2], or duromeric polymer materials such as epoxies and polyurethanes. Using foams results in a decrease in the total mass of material required and is advantageous in many applications, e.g., for roof design. Polymeric foams, in particular due to the low density of polymers themselves, lead to the lowest loading on a flat roof for a given target thermal transmission and load-bearing. Another major aspect of designing roofs is the need to ensure flame retardancy. All materials for buildings have to be flame-retarded, difficult to ignite, and with low fire load [3,4]. The respective tests include the vertical flame spread (VTS) in Europe, the VTS combined with the limiting oxygen index (LOI) in China, the tunnel test in the US, and the cone calorimeter test in Japan. Only the tests according to ISO 1182 (EU) and BS 5852:2006 (GB, for furniture) take the mass loss as a specification into account. However, this residue is crucial: the thermal insulant has to maintain its structure in a fire to protect steel in reinforced structures from weakening. Foamed thermal insulation materials therefore should not melt, making crosslinked polymer foams ideal candidates. Rigid polyisocyanurate (PIR) foams are characterized by crosslinking via isocyanurate rings which are generated by trimerization of an excess of isocyanate molecules in the foaming process. Isocyanurate groups have higher thermal stability than other polymeric building blocks [5,6] and are the main source of generated char. PIR structures are very rigid. In the past, this was compensated by chlorofluorocarbon (CFC) blowing agents that, together with tris(chloroalkyl)phosphate flame retardants (FRs) [6], served as a plasticizer.

PIR foams are now the dominant duromeric foam insulants in European standard applications. They still use the highly effective tris(chloroisopropyl)phosphate [7] today, with pentane as a blowing agent. However, halogen-free PIR technologies are on their way, and triethyl phosphate (TEP) seems to be emerging as a halogen-free substitute [8]. If TEP alone does not induce a sufficient effect or is excessive in its plasticizing effect, triaryl phosphates such as triphenyl phosphate (TPP) may be additionally added [9]. These FRs are nonreactive FR additives [7]. It has to be noted that environmental concerns about such compounds have been raised [10]. The often better solution is to incorporate the FR into the polymer backbone to avoid migration. This concept is well-established in the academic literature also for polyurethanes [7,11]. Halogen-free solutions in practice are Exolit® OP550 (a phosphate-containing polyol) and OP560 (a phosphonate-containing polyol), designed for flexible polyurethane foams. Recently, diethylhydroxymethylphosphonate (DEHP) has been launched into the market by two suppliers, and dibutylhydroxymethylphosphonate (DBHP) has been proposed as a variant with better plasticizing performance [12].

Although a number of halogen-free FRs are available, there is still a lack of systematic research. The aim of this contribution is to present the results of systematic studies for phosphorus-containing FRs in PIR foams. The research concept is illustrated in Figure 1. With the aim to compare the influence of different chemical structures, an aromatic phosphinate (9,10-dihydro-9-oxy-10-phosphaphenantrene-10-oxide, DOPO) and an aromatic *H*-phosphonate (dibenzo[d,f][1,3,2]dioxaphosphepine 6-oxide, BPPO) were chosen. Both phosphinates and *H*-phosphonates can be employed in phospha-Michael addition reactions, as reviewed extensively by Salmeia et al. [13] and by Döring et al. [14,15,16,17,18]. The reaction of double bonds with either DOPO or BPPO leads to nonreactive FR additives (left-hand side of Figure 1). New phosphonate-based FR additives for rigid PIR foams containing BPPO rings were reported by our group. The *H*-phosphonate BPPO was reacted with acrylates and benzoquinone, respectively [19]. The Pudovik or phospha-aldol reaction of DOPO and BPPO with aldehydes (right-hand side of Figure 1) [20,21,22,23] generates an aliphatic OH group that can react with isocyanate during the polyaddition reactions of the PIR foam. These OH groups are chain terminators forming dangling ends of the polymer network. Dialdehydes yield two aliphatic OH groups able to be fully incorporated into the PIR polymer. Phosphinate/aldehyde reactions have been reported for the reaction of diphenylphosphite with benzaldehyde (BA) [23], acetaldehyde [24], formaldehyde, and terephthaldialdehyde [25,26], but not with furfural. Reactions of aliphatic and aromatic *H*-phosphonates with benzaldehyde in presence of catalysts (acid, base, or metal fluoride) were reported by Fang et al. [21]; the reaction catalyzed by n-butyl lithium as strong base was described by Liu et al. [27].

The influence of the nonreactive and reactive P-containing FRs for PIR foams was studied and compared to the present state of the art. Biogenic educts such as furfural (one of the top chemicals from biorefinery carbohydrates [28]), benzaldehyde, and terephthalaldehyde were employed to enhance the sustainability of the reactive FRs. The efficacy of the different FRs was compared with PIR foams without FR, with TEP, TPP, with a mixture of TPP and TEP, and with the phosphonate-containing diol Exolit^®^ OP560. These foams had to have roughly comparable physical properties (i.e., density, open porosity, cell morphology) to achieve reliable results. Therefore, the parameters of foam formulation and for the foaming procedure were kept as constant as possible. The FRs were incorporated into PIR foams with constant formulation with a total P-content of 0.7 wt.% (FR alone) or plasticized with TEP at a total P concentration of 1 wt.%. The thermal decomposition was examined by thermogravimetry. The flame and fire behaviors of the foams were measured by the vertical flame spread test and forced flaming combustion in the cone calorimeter. The data concerning total smoke release (TSR), CO, and CO_2_ production are also reported. Besides fire-relevant parameters, the morphology of the resulting char residues, being crucial for protecting the underlying material, was studied. Here, we were interested, in particular, in the influence of the different FR additives on the charring.

## 2. Materials and Methods

### 2.1. Chemicals

Paraformaldehyde (>95.0%, Sigma-Aldrich, Darmstadt, Germany); furfural (furane-2-carbaldehyde, >99.0%, Sigma-Aldrich, Darmstadt, Germany); benzaldehyde (>99.0%, Sigma-Aldrich, Darmstadt, Germany); diethyl phosphonate (>97.0%, Sigma-Aldrich, Darmstadt, Germany); dibutyl phosphonate (>98.0%, Sigma-Aldrich, Darmstadt, Germany); terephthalaldehyde (>99.0%, Sigma-Aldrich, Darmstadt, Germany); pentane (≥99.0%, Sigma-Aldrich, Darmstadt, Germany); triphenyl phosphate (TPP, >99.0%, TCI, Eschborn, Germany); triethyl phosphate (TEP, >99.0%, TCI, Eschborn, Germany); poly(ethylene glycol) (PEG 400, *n_OH_* = 280 mg KOH⋅g^−1^, Sigma-Aldrich, Darmstadt, Germany); potassium acetate (KAc, ≥99.0%, Sigma-Aldrich, Darmstadt, Germany); diethylene glycol (DEG, >99.0%, Sigma-Aldrich, Darmstadt, Germany); phthalate/DEG polyester polyol Desmophen^®^ PEP50AD (*n_OH_* = 240 mg KOH⋅g^−1^, Covestro Deutschland AG, Dormagen, Germany); polymeric isocyanate DESMODUR^®^ 44V70L (*c_NCO_* = 30.9%, Covestro Deutschland AG, Dormagen, Germany); Emulsogen^®^ TS100 (Clariant, Pratteln, Switzerland); polyether-modified polysiloxane TEGOSTAB^®^ B 8421 (*n_OH_* = 57 mg KOH⋅g^−1^) (Evonik Industries, Darmstadt, Germany); Sacoflam 47201 (AC4-DOPO, Metadyna, Austria); Exolit^®^ OP 560 (Clariant, Hürth, Germany); p-tolylisocyanate (99%, Sigma-Aldrich, Darmstadt, Germany); acetaldehyde (>99.0%, Sigma-Aldrich, Darmstadt, Germany); triethylamine (99.0%, Sigma-Aldrich, Darmstadt, Germany) were used as received. For PIR foam preparation, a solution of 25 wt.% potassium acetate (KAc) in DEG (*n_OH_ =* 793 mg KOH⋅g^−1^) was used as a catalyst.

### 2.2. Synthesis of Flame-Retardant Additives

#### 2.2.1. Phospha-Michael Additions

The synthesis of phospha-Michael adducts by reactions of acrylates (BuA: butylacrylate; DMI: dimethylitaconate; EA: ethylacrylate; PA: phenylacrylate) and benzoquinone with BPPO and DOPO, respectively, with all experimental details was published by our group [19,29].

#### 2.2.2. Phospha-Aldol Reactions

The chemical structure and the abbreviations of the BPPO- and DOPO-adducts with aldehydes as well as P-content, yield of the synthesis, and molecular weight are summarized in Table 1.

The synthesis of the phospha-aldol adducts is described in the following. 

The procedure for preparation of FA-BPPO (**1a**) as given by Wang et al. [30] was slightly modified. Thus, BPPO (30.0 g, 0.13 mol) and paraformaldehyde (3.88 g, 0.13 mol) were dissolved in dry tetrahydrofuran (50 mL) and refluxed for 3 h. At the end of the reaction, after cooling, the raw product was precipitated, isolated, washed with diethyl ether, and dried.

The synthesis of AA-BPPO (**2a**), FU-BPPO (**3a**), and BA-BPPO (**4a**) was performed according to Kumaraswamy et al. [31] with slight modifications. BPPO (30.00 g, 0.13 mol) and aldehyde (0.13 mol) were dissolved in dry dichloromethane (50 mL), and the mixture was cooled with ice. Triethylamine (0.3 mL, 0.5 mol% based on BPPO) was added, and the reaction mixture was stirred at 0 °C for 2 h. At the end of the reaction, the precipitate was isolated, washed with cold dichloromethane, and dried.

TA-BPPO (**5a**) was prepared according to the instructions of Cao et al. [25] with slight modifications. BPPO (250.0 g, 1.08 mol) and terephthalaldehyde (72.0 g, 0.54 mol) were dissolved in dry toluene (500 mL) and stirred for 6 h at 120 °C. At the end of the reaction, the precipitate was filtrated, washed with diethyl ether, and dried.

The addition of DOPO to the corresponding aldehydes (**1b**–**5b**) was performed in analogy to the synthesis of the BPPO-aldehyde derivatives.

The ^1^H, ^13^C, and ^31^P NMR data and spectra of all compounds with signal assignments are given in the Appendix A (Appendix A). The analyses confirmed the structure of all target compounds.

### 2.3. Foam Preparation

The general formulation for the preparation of foams is given in Table 2. An NCO/OH molar ratio of 3.2 (NCO index 320, typical for steel-faced sandwich panels with a core of PIR foam) was kept constant for all the foams. At this NCO index, the foams contain a mixture of urethane and isocyanurate structures, as reported previously [32]. For simplicity, the foams are referred to as PIR foams. The composition of all PIR foams investigated in the study is summarized in Appendix A.

### 2.4. Foam Properties

The density, pore sizes, and cell integrity of the foams were characterized. The samples used for cone calorimetry (10 cm × 10 cm × 5 cm) were employed to calculate the density from volume and weight. Pore sizes were determined from light microscopic images. The microscope used was an Axio Imager (ZEISS) equipped with Axiocam 305 color camera (ZEISS, Oberkochen, Germany).

The scanning electron microscopy (SEM) images were taken by a Gemini Ultra plus SEM (ZEISS, Oberkochen, Germany). Water absorption was determined as a parameter for the cell integrity. Foam cubes of 4 cm × 4 cm × 4 cm were completely immersed in boiling water for 90 min. The mass difference (*m_w_* − *m_d_*) was determined (*m_w_* describes the mass of the wet foam, *m_d_* mass of the dry foam). For the density of water (*ρ* = 1 g·cm^−3^), the mass difference was divided by the sample volume (*a*^3^) and normalized to 100%. The water absorption *WA_V_* (volume of absorbed water over foam volume) was obtained with Equation (1).
(1)$$WA_{V}=\frac{m_{w}-m_{d}}{a^{3}\times \rho }\times 100\%$$

Foams with $WA_{V}<20\%$ were designated as closed-cell, foams with $WA_{V}>20\%$ as open-cell foams.

### 2.5. Chemical Analysis

#### 2.5.1. Nuclear Magnetic Resonance Spectroscopy

^1^H, ^13^C, and ^31^P NMR spectra were recorded using a Bruker Avance III 500 NMR spectrometer (Germany) operating at 500 MHz for ^1^H, at 125 MHz for ^13^C, and at 202 MHz for ^31^P NMR, respectively. DMSO-d_6_ was used as solvent and internal standard (δ(^1^H) = 2.50 ppm, δ (^13^C) = 39.6 ppm). The ^31^P NMR spectra were referenced on external H_3_PO_4_. The signal assignments are based on the evaluation of 1D and 2D NMR experiments.

#### 2.5.2. Raman Spectroscopy

The Raman spectra were acquired with a Raman Imaging System WITEC alpha300R (Ulm, Germany) with a laser wavelength of 532 nm. The laser power was 500 μW. The spectra were smoothened using the Savitzky–Golay method.

#### 2.5.3. Elemental Analysis

The quantitative determination of the elements hydrogen, carbon, and nitrogen was carried out by means of elementary analysis. The analyses were performed with a MICRO CHNS Elemental Analysis (Elementar Analysensysteme GmbH, Hanau, Germany).

#### 2.5.4. Quantitative Phosphorus Content

The quantitative phosphorus content was determined by Mikroanalytisches Labor Kolbe (Oberhausen, Germany).

#### 2.5.5. Titration of OH Groups

The OH group titration was performed with a Mettler T70 titrator with DG-111 sensor (Giessen, Germany). The determination was performed according to DIN 53240-2. The acetylation mixture was an acetic anhydride solution with 10 vol% N-methylpyrrolidone (NMP). The catalyst solution consisted of a solution of 4-N-dimethylaminopyridine (1 wt.%) in NMP. After acetylation of the sample, the acetic acid released from acetic anhydride was titrated with 0.5 N KOH in methanol.

### 2.6. Thermal Decomposition Behavior

The thermal decomposition was assessed by thermogravimetric analysis (TGA) using a TGA Q500 (TA Instruments, New Castle, UK) in nitrogen atmosphere (60 mL·min^−1^) in the temperature range of 25 to 800 °C at a scan rate of 10 K·min^−1^ with a sample weight of 5 mg. The thermal decomposition products were analyzed with pyrolysis–gas chromatography coupled with mass spectrometry (GC/MS). These experiments were carried out with a GC 5890 (Agilent Technologies, Santa Clara, CA, USA) coupled with a pyroprobe 2000 (CDS Instruments, Oxford, MA, USA) under helium atmosphere and a flow rate of 1.0 mL·min^−1^.

### 2.7. Fire Behavior

#### 2.7.1. Vertical Flame Spread (VFS)

The vertical flame spread test was carried out according to DIN 4102. Samples with dimensions of 20 cm × 10 cm × 1 cm were vertically attached in the test chamber (here, a test chamber for UL-94 test was used). A burner flame was applied for 15 s on the lower edge of the specimen. The height of the flame was measured visually. 

#### 2.7.2. Cone Calorimeter Test (Forced Flaming Combustion)

The PIR foams were cut into specimens with dimensions of 10 cm × 10 cm × 5 cm. The samples were conditioned at 23 °C for 48 h in a climate chamber with 50% relative humidity before fire testing. They were tested in a cone calorimeter (Fire Testing Technology, East Grinstead, UK) according to ISO 5660-1. The heat flux was adjusted to 50 kW·m^−2^. The distance of cone heater to sample was 25 mm. Aluminum foil was wrapped around the sides of the probe to avoid edge burning. During the measurement, time to ignition (t_ig_), heat release rate (HRR), peak of heat release rate (PHRR), time to PHRR (t_PHRR_), maximum of the average rate of heat emission (MARHE), total heat released (THR), total mass loss (TML), and effective heat of combustion (EHC, calculated by THR/TML) were evaluated.

#### 2.7.3. Morphology of the Remaining Chars

The chars remaining after cone calorimetric investigation were inspected visually and by SEM as outlined in Section 2.4. Photographic images of the chars were taken as side view to measure the height.

## 3. Results and Discussion

### 3.1. Synthesis of Phospha-Michael and Phospha-Aldol Derivatives of BPPO and DOPO

The synthesis of the phospha-Michael adducts of BPPO to acrylates and benzoquinone has been reported before by our group [19]. All nonreactive FRs employed here have been described there as well. The reaction of *H*-phosphonates and phosphinates with aldehydes follows the mechanism of a Pudovik reaction: a base-catalyzed addition of a carbonyl to a P-H-acidic compound [22]. This reaction generates phosphorus-carbon linkages as the phospha-Michael addition, with the difference that the resulting compound bears an aliphatic hydroxyl group. In the first step, a five-membered ring of phosphonate, aldehyde, and triethylamine is formed. In this transition state, the triethylamine first binds the hydroxyl proton of phosphonate. The subsequent bond rearrangement and cleavage of triethylamine will give the desired product. A summary of the phospha-aldol adducts of DOPO and BPPO to aldehydes used in this study is given in Table 1. The yields were in the range of 55–89%, and the purity was high according to the ^1^H and ^31^P NMR spectra (see Appendix A, Appendix A).

### 3.2. Characteristic Parameters of the PIR Foams Prepared under Comparable Conditions

Nonreactive and NCO-reactive BPPO- and DOPO-containing FRs as well as controls with TEP, TPP, a mixture of TPP and TEP, and Exolit^®^ OP 560 were used in the typical PIR foam formulation according to Table 2. The formulations without TEP aimed at a final P-content of 0.7 wt.%, while a P-content of 1.0 wt.% was targeted after plasticizing with TEP. The most important parameters of the formulation (NCO/OH molar ratio = 3.2) and the catalyst KAc concentration to control the kinetics of the reaction were kept constant, as it was shown before that the catalyst concentration had a tremendous effect on foam composition and its fire-retardance properties [32]. A summary of the most important samples employed for the discussion is given in Table 3. The compositions of all foams studied are summarized in the Appendix A (Appendix A).

PIR foams with densities in the range of 38 ± 3 kg⋅m^−3^ (i.e., most of the foam samples) can be compared in their properties without any problem. Samples with densities outside this range should be compared with care. PIR foams reported by Günther et al. [33] and Lorenzetti et al. [34] also showed compression strengths of approximately 320 kPa at NCO indexes and densities comparable to this study. Density and compression strength correlated (see Appendix A): the higher the density, the higher was the compression strength of the samples (Appendix A). All compression strengths measured are given in Appendix A.

A complete solubility of the FRs in the formulation was not observed since the foam formulations turned opaque under stirring. The formulation temperature rose to T ~ 120 °C. This temperature increase enhanced the solubility of the additives and supported the generation of more homogeneous foams with closed-cell morphology. The melting points of the phospha-Michael adducts were in the range of 76−125 °C [19] and melted within the generating foam. Consequently, these adducts yielded homogeneous morphologies. The melting points of phospha-aldol adducts were higher, in the range of 143−234 °C (see Table 1), and therefore, their solubility in the formulation was less pronounced. Thus, these additives yielded heterogeneous morphologies with (mostly) an open-cell character (see Figure 2). The solid particles with melting points above the foam temperature also affected the stability of the cell walls, resulting in a loss of blowing agent during foaming and in a higher density of the foam.

The average cell size as determined visually in the SEM images ranged typically at 0.18 ± 0.04 mm. Only the foams with EA-BPPO-0.7, BA-BPPO/TEP-1.0, and TA-BP/TEP-1.0 were coarser-celled. 

The water uptake *WA_v_* can be regarded as a measure for the content of open cells in the foams. The values given in Table 3 show that the water uptake *WA_v_* of the control samples as well as the foams with the nonreactive FRs was rather low, between 4–16%. The water uptakes of the foams with reactive FRs were significantly higher, in particular for DOPO-based additives (in the range of 90%), indicating open-cell morphologies. NCO-reactive BPPO-derivatives also resulted mostly in open-cell structures with *WA_v_* between 50−67%. Addition of TEP to the foams often enhanced the open-cell content because cell walls became softer (compare sample FA-BPPO-0.7 with *WA_v_* = 67% with sample FA-BPPO/TEP-1.0 with *WA_v_* = 82%). There were just a few examples of foams containing FRs with melting points > 100 °C that were closed-cell. This supports the hypothesis that nonmolten particles led to cell rupture. In three of the four cases where closed-cell foams contained NCO-reactive FRs with melting points > 100 °C, the reaction of isocyanate with the FR seems to have provided the necessary dispersion of FR in the matrix to avoid cell wall rupture. The question of why this happens in some cases and not in all cases with high-melting NCO-reactive FRs remains open for future investigations. The results of the water uptake strongly indicated the difference between open-cell for phospha-Michael derivatives (see also [29,32]) and closed-cell (mostly phospha-aldol) foams. This was verified by SEM, as illustrated for selected examples in Figure 2. Open-cell morphologies appear more irregular than closed-cell morphologies.

### 3.3. Thermal Decomposition and Pyrolysis

As the focus of this study was on the fire behavior of the PIR foams, the thermal decomposition behavior is summarized only briefly. The data relevant for the discussion are given in detail in the Appendix A (Appendix A). The thermal decomposition of the PIR foams was studied by TGA under nitrogen between 30 and 800 °C. TGA curves of selected samples of each series are displayed in Figure 3. 

The curve of the pure PIR foam without FR additive (Figure 3a) shows four decomposition maxima ((1)–(4)). The main decomposition maximum at T = 320 °C originates from the decomposition of the polyester polyol and isocyanate and is designated with (1). At this temperature, pyrolysis-GC/MS (py-GC/MS) detected fragments of the polyesterpolyol PEG 400, such as DEG, and often also higher ethylene glycols such as tri- and tetraethylene glycol (at higher temperature deliberating dioxane), phthalic acid derivatives, and aminotoluene, benzylaniline, and aniline as decomposition products of the polyisocyanate. The CO_2_ originating from urethane decarboxylation was not detectable. The py-GC/MS data are summarized in the Appendix A (Appendix A) and in ref. [35].

Addition of the phosphorus-containing FRs (both phospha-Michael as well as phospha-aldol adducts) did not change the position of the maxima (1)–(4) significantly, as illustrated by the TGA curve of the foam with the Pudovik adduct of formaldehyde to butylphosphonate (FA-BP). Addition of TEP resulted in new maxima in the TGA curves designated ((**5a**) and (**5b**)) in the temperature range 170−233 °C (Figure 3a). These maxima were found in all TEP-containing PIR foams (compare the summary of main signals observed in py-GC/MS investigations in the Appendix A, Appendix A). Py-GC/MS proved that the maxima were indeed caused by the evaporation of TEP [29]. TEP evolved at higher temperature (324 °C), too. This was mainly observed in PIR foams containing reactive FRs (phospha-aldol adducts). Comparing the TGA curves of FU-BPPO and FU-DOPO without and with TEP (Figure 3b), it can be noticed that TEP did not change the decomposition significantly. Obviously, TEP did not interfere with the decomposition of the PIR foam and just acted as a gas-phase FR agent. The furfural adducts FU-BPPO and FU-DOPO caused generation of furfural at T = 405 °C, for FU-BPPO together with 2,2′-biphenyl diol, which is a clear indication of the decomposition of FU-BPPO. DOPO derivatives generally enhanced the intensity of the decomposition at maximum (2) and shifted it to slightly higher temperature of around 425 °C. Py-GC/MS at that temperature revealed the occurrence of a number of mass fragments, e.g., with m/z = 31 (referring to P) and m/z = 147 (dibenzofurane), which can be assigned to DOPO fragments [18]. The decomposition of DOPO and BPPO differed significantly because the intramolecular substituent cleavage from the former seems to be much easier than hydrolytic separation from the latter. While the ^31^P-NMR shifts (see Appendix A) do not justify differentiating DOPO and BPPO relating to their electron density or “formal oxidation state” (P(V)), the kinetics of substituent extrusion and liberation of active phosphorous species into the gas phase may play a crucial role in understanding subtle differences between the FRs. 

Summarized at this point, the thermal decomposition of PIR is only limitedly altered by the phosphorus-containing FRs. They generally moderately enhanced the residues of the samples obtained after TGA at 800 °C from 26.6 wt.% for PIR-0 to about 30−33 wt.% for reactive phospha-aldol BPPO adducts (FU-BPPO/TEP-1.0; TA-BPPO/TEP-1.0; FU-BP/TEP-1.0). All the observations indicated that a combined gas phase and condensed phase mechanism take place during thermal decomposition.

### 3.4. Fire Behavior

The main task of this study focused on the fire behavior of the PIR foams modified with halogen-free nonreactive and reactive FRs in comparison to state-of-the-art solutions. The VTS test served as the first indication to qualify the influence of the FRs. In one of our former reports [29], the VTS correlated well with peak of heat release rate (PHRR) values obtained by forced flaming combustion in the cone calorimeter. The residues remaining after the cone calorimeter tests were examined to get an impression of the structure and composition of the chars after fire, which is of important practical relevance.

#### 3.4.1. Vertical Flame Spread (VFS)

The VFS test, according to DIN 4102, can be considered as the first measure for the evaluation of burning behavior. The flame height, given in cm, decides about the classification into the categories “B2 classification passed” (satisfactory properties, flame height ≤ 15 cm) or “not passed” (flame heights > 15 cm). The vertical flame spread is known to be sensitive to the cell size, the content of the cells, and the immediate availability of gas-phase active FR. The flame retardant has, as shown in Table 3, an impact on the water uptake of the foam, which reflects the content of closed cells. We considered foams with water uptake < 20% as “closed-cell” and foams with a water uptake of > 20% as “open-cell”. This is assumed to play a role for the content of pentane and air in the cells. Control and reference foams were all closed-cell. The control foam PIR-0 without P burned completely. The controls TEP-0.3 gave a VFS of 18 cm and TPP-0.7 15 cm, and the combination in the state-of-the-art PIR TPP/TEP-1.0 yielded a VFS of 14 cm. The results of the small burner test with VFS of the PIR foams with phosphorus-containing FRs are given in Table 4 and Table 5. The values illustrate that the chemical nature of the FR molecule impacts the foams’ FR performance. Foams with nonreactive phospha-Michael adducts at a phosphorus content of 1 wt.% usually exhibited VFS >15 cm, as reported before [29]. The addition of TEP resulted in a reduction of VTS by about one unit in most combinations. The reactive phospha-aldol FRs based on aliphatic aldehydes resulted in foams with unsatisfactory VFS (flame height > 15 cm). If aromatic aldehydes were used, very low flame heights (<10 cm, 11cm) were observed, indicating very good flame-retardant properties. DOPO derivatives performed slightly better than BPPO derivatives. The best results (VFS < 10) were achieved when the difunctional phospha-aldol adducts of BPPO, DOPO, and BP, respectively, to aromatic terephthalaldehyde were built into the polymer chain and the FR did not act as a chain terminator. This supports our previous conclusion that crosslink density and FR performance tend to correlate [32]. The results with biobased furfural adducts to DOPO and BPPO combined with TEP also indicated efficient flame retardants in the PIR foams, but their flame heights were slightly higher (12−14 cm) than for the TA adducts.

#### 3.4.2. Forced Flaming Combustion (Cone Calorimeter)

Forced flaming combustion studies in the cone calorimeter were carried out to obtain deeper insights into the fire behavior of the PIR foams. The cone calorimeter test has been established as a scientific tool to investigate the burning of materials under the typical conditions of a developing fire scenario [33,36]. Valuable information can be derived, e.g., from the heat release curves, as illustrated in Figure 4 for representative samples of each group (controls, nonreactive FR, reactive FR). First, the control foams and PIR foams with the state-of-the-art FRs were examined. Then, the influence of nonreactive and reactive FRs based on BPPO, DOPO, and BP was studied with respect to the fire-relevant parameters and afterwards with respect to the structure and morphology of the remaining chars. All relevant parameters for the samples discussed in the following are summarized in Table 6 and Table 7.

.

The time to ignition was not influenced by the FRs and remained constant between one and two seconds. This is due to the excellent thermal insulating properties of the foams: dissipation of heat is initially not possible. The time to PHRR ranged, also almost not influenced, between 9 s and 13 s (typically between 9−10 s). The rapid flame propagation was followed by the formation of a carbonaceous layer causing a fast drop in heat release rate, which was not observed as clearly in the control foam without P. The carbonaceous layer functioned as a protective layer. Thus, an increase in surface temperature yielded an efficient heat shielding effect, reducing the heat release rate [8,33].

Even the cell walls carbonized and stood intact, so that the morphology of the fire residue supported the generation of a temperature gradient between surface and pyrolysis front via thermal insulation. After about 50 s, a roughly constant HRR was reached that remained stable until a burning time of about 200 s. After that time, the HRR decreased further until the end of the test. In some experiments, for instance with the PIR foams with TPP/TEP, FA-BP, and FA/TA-BPPO, a second maximum was observed between 125 and 200 s burning time. This can be explained by the fracture of the carbonaceous layer and the generation of new surfaces, providing more fuel. This supports the hypothesis that the mechanical stability of the remaining char can be of high importance for fire safety.

Table 6 displays the fire-relevant parameters of control and reference samples. The control foam PIR-0 had a MARHE value of 172 kW⋅m^−2^. This value was reduced effectively by both TEP and TPP. TEP (0.3 wt.%) enhanced the char yield only slightly from 22 to 27 wt.%, while TPP (0.7 wt.%) caused a significant enhancement to 45 wt.%.

The maximum average rate of heat emission (MARHE) was plotted vs. the remaining residue (char yield) as one measure for the fire performance of the FRs. These two parameters take two important processes of a fire into account: the heat emission, which should be reduced, as well as the char formation, which should be enhanced by the FR. The plot for the control samples given in Appendix A (Appendix A) illustrates that addition of a mixture of TPP and TEP introduced an additive effect into the fire behavior.

In the next step, TPP was substituted subsequently by the BPPO-, DOPO-, and BP-based FRs. Nonreactive BPPO and DOPO adducts (EA-BPPO/TEP-1.0, DMI-BPPO/TEP-1.0, DMI-DOPO/TEP-1.0, and PA-BPPO/TEP-1.0) decreased MARHE efficiently without significant differences. A slightly better performance of aromatic compared to aliphatic FRs was noticed. This was accompanied by moderate enhancement of char (Appendix A). The DOPO derivative DMI-DOPO caused the highest increase in char yield, which is assumed to be originated by the different combustion of DOPO compared to BPPO and the higher condensed phase action [18]. Note that the large residue with BuA-DOPO was caused by incomplete pyrolysis of the sample due to the formation of an effective carbonaceous protecting layer.

The influences of the chemical structure of phospha-aldol derivatives using the formaldehyde adducts of different phosphorus compounds as FR is illustrated in Figure 5a. All of them had a chemical linkage to the PIR foam via one OH group, but the phosphorus is contained in an aromatic phosphonate (BPPO), an aliphatic phosphonate (BP), or a phosphinate (DOPO). FA-DOPO/TEP showed the best performance in the PIR foam with low MARHE, combined with moderate char yield increase, while FA-BPPO/TEP did not reveal a significant effect. Aliphatic-aromatic phosphonates (FA-BP/TEP) had values comparable to the TPP/TEP control.

The effect of the chemical structure of the aldehyde in BPPO adducts combined with TEP is outlined in Figure 5b. Both an aromatic structure as well as an increasing number of OH groups improved the FR performance. The best value for MARHE and char was achieved for the terephthalaldehyde adducts. It had to be noticed that the char remaining from these foams was very fragile and mechanically not stable (see discussion later in the section). 

It can be noted at this point that the best FR performance can be ascribed to FU-BPPO/TEP.

Forced flaming combustion in the cone calorimeter resulted in all samples in the formation of significant char, mostly with defined geometry, which can be expressed by the height of the char and the amount of char (both can be found in summarizing Table 7 and Table 8). While PIR-0 yielded a condensed char with low height, addition of all different P-containing FRs mostly defined chars with heights comparable to the nonburned sample (5 cm). Only the foams with TA-BPPO/TEP and FA-BPPO/TEP were mechanically less stable. TPP clearly stabilized the char mechanically, which is caused by the different burning behavior due to the phosphate structure. The same stabilizing effect was observed for FA-DOPO/TEP, FU-DOPO-TEP, and TA-DOPO/TEP. The stabilizing effect with DOPO derivatives is ascribed to condensation reactions of the phosphinate reaction products in the char, as found before [18]. 

Table 9 shows photographical images of the remaining chars obtained after cone calorimeter study of selected samples, as mentioned before.

The morphology of the remaining chars of selected samples visualized by SEM is shown in Figure 6a–e. The images of the chars without any FR additive, with TPP/TEP, and with FA derivatives illustrate a rather smooth surface (Figure 6a–e), while addition of BuA-DOPO/TEP and FU-BPPO/TEP yielded finer char morphologies with probably higher mechanical stability and proper cohesion in the char (Figure 6f,g).

The phosphorus content of selected chars was determined by elemental analysis, and it was found that the examined samples showed significant contents of P after burning (Table 6 and Table 7). The highest P contents (between 0.90 and 1.50 wt.%) were found in burned PIR foams with EA-BPPO, FU-BPPO/TEP, and FU-DOPO. The high content with BuA-DOPO was due to incomplete burning of the samples.

All the results obtained so far clearly indicated a combined gas phase and condensed phase action of the phospha-Michael and phospha-aldol adducts.

Smoke production and CO production are influenced by several factors. Phosphorus in the condensed phase may enhance charring [37], but it also increases the production of volatile aromatic precursors for smoke. An efficient charring and a complete protective residual layer reduces the release of any kind of fuel and results thus in a drastic reduction in CO and smoke production. Flame inhibition by phosphorus in the gas phase is believed to increase the smoke production and to stabilize soot against oxidation [38]. Additionally, the CO yield is said to be increased independent of the fact that this is due to the reduction of flame temperature or due to radical scavenging. Total smoke release (TSR) and CO yield monitored in the cone calorimeter investigation were considered in order to get an impression of the influence of the P-containing FRs on the gas-phase behavior during fire. The CO yield (values given in Table 6 and Table 7) was taken as one aspect and measure of smoke toxicity [4,39].

The control foam PIR-0 produced a CO value of 0.04 kg⋅kg^−1^. This CO yield was barely increased by TEP but significantly by TPP (0.10 kg⋅kg^−1^). Addition of phospha-Michael derivatives caused a significant rise to values between 0.06 to 0.14 kg⋅kg^−1^. The highest values for the CO yield were obtained with BuA-DOPO as FR. On the other hand, the systems with BuA-DOPO had the lowest mass loss, compensating the increase in CO yield to a large extent with respect to the absolute CO production.

Among the phospha-Michael derivatives, both DMI-BPPO and DMI-DOPO appeared optimal due to the low CO yields compared with low MARHE values. However, it is clearly visible that NCO-reactive phospha-aldol derivatives had an even better performance. Their addition resulted in lower CO yields (0.05⋅0.08 kg⋅kg^−1^) combined with low MARHE. This illustrates the positive effect of the chemical linkage between the FR and the PIR network. Exceptions were the TA- and FU-BP derivatives which are aliphatic phosphonates with easily cleavable groups. The addition of these FRs induced strong CO formation (0.18 kg⋅kg^−1^). Even higher values were reached with the aliphatic Exolit^®^ OP 560 (0.20 kg⋅kg^−1^), although the diol was completely trapped in the network. It can be concluded that Exolit® OP 560 is not an optimal FR for rigid PIR foams regarding all aspects (high MARHE, high smoke release, and CO yield).

## 4. Conclusions

In this study, the influence of phosphorus-containing FRs with systematically altered chemical structure on the thermal decomposition and fire behavior of rigid PIR foams was studied by a combination of suitable methods. As basic phosphorus-containing molecules, the aliphatic phosphonate BP, the aromatic phosphonate BPPO, and the phosphinate DOPO were employed. They were incorporated either as nonreactive additives (phospha-Michael adducts) or as NCO-reactive additives (phospha-aldol adducts) into the PIR foams with comparable chemical composition and physical properties. The schematic structures of the resulting foams and their impact on the flame and fire-retardant properties are summarized and illustrated in Table 10. 

FRs without any NCO-reactive group were dissolved and/or evenly distributed within the PIR matrix due to the fact that their lower melting points supporting dissolution in the foam formulations. The OH groups of the phospha-aldol adducts were proven to react with the isocyanate groups in the formulations. One NCO-reactive OH group resulted either in chain termination of the PIR network segments or in dangling chains. Two reactive OH groups formed segments where the phosphonate or phosphinate group was dangling to an intact polymer chain of the PIR network. For comparison, an aliphatic, P-containing diol was employed (Exolit^®^ OP 560).

The studies of the forced-flaming combustion in a cone calorimeter demonstrated that the type of morphology significantly determined the burning behavior.

The subtle changes in the chemistry around the P atom caused significant changes in the FR effect with respect to both heat release and charring. Most of the phosphinate DOPO derivatives performed better than phosphonate BPPO adducts, with few exceptions. One reason, therefore, is seen in the different thermal decomposition of DOPO and BPPO. DOPO decomposes through intramolecular extrusion of dibenzofurane. This process appears to be much faster than the scission of bisphenol from BPPO.

Reactive FRs had a higher impact than nonreactive FRs. The aromatic BPPO adducts with two OH groups (TA-BPPO) induced the lowest MARHE but also an instable, highly compact char. An optimum effect was recognized for PIR foams with FU-BPPO and FU-DOPO with one OH group. The choice of the furfural derivatives is a first step toward a renewable materials base for PIR foams.

Fire is a fast process, and thus, the differences in FR action are dominated by specific interactions during pyrolysis and decomposition kinetics, dispersion, particle surface/mole ratio, melting point, and volatility. Although the main flame-retardant modes of action such as charring, flame inhibition, and protective layer are rather well-known, the specific mechanisms determine that flame retardants are very different when compared systematically in detail.

The investigations of the fire behavior of rigid PIR foams with BP, BPPO, and DOPO adducts showed that effective, though expensive, flame retardants are available to replace TCPP with comparable results to benchmark foams. The fine-tuning of the chemical structure should be directed to the specific applications of the PIR foam. The results also showed that incorporation of the FR unit as dangling chain by a thermally labile chemical bond is more favored than incorporation as chain terminator. The molecules developed here can also have potential applications as FRs in other polymers, such as polyamide or polyester resins.

## Figures and Tables

**Figure 1 materials-16-00172-f001:**
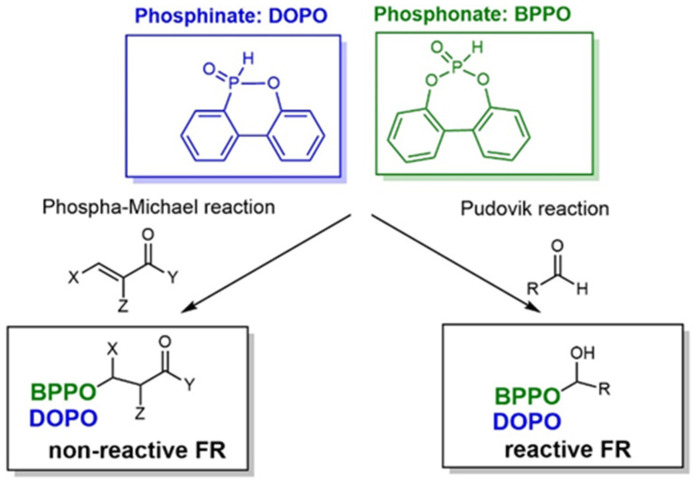
Research concept for the development of phosphorus-containing halogen-free FRs for PIR foams.

**Figure 2 materials-16-00172-f002:**
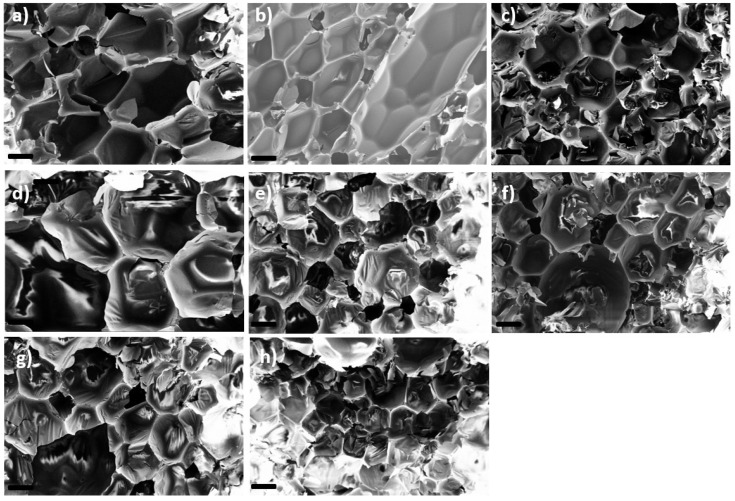
SEM images of selected PIR foams with closed-cell and open-cell morphology (scale bar: 200 µm): (**a**) PIR-0 (closed-cell); (**b**) TPP/TEP-1.0 (closed-cell); (**c**) EA-BPPO/TEP-1.0 (closed-cell); (**d**) FA-BPPO/TEP-1.0 (open-cell); (**e**) FU-BPPO/TEP-1.0 (open-cell); (**f**) FU-DOPO/TEP-1.0 (closed-cell); (**g**) TA-BPPO/TEP-1.0 (open-cell); and (**h**) TA-DOPO/TEP-1.0 (open-cell).

**Figure 3 materials-16-00172-f003:**
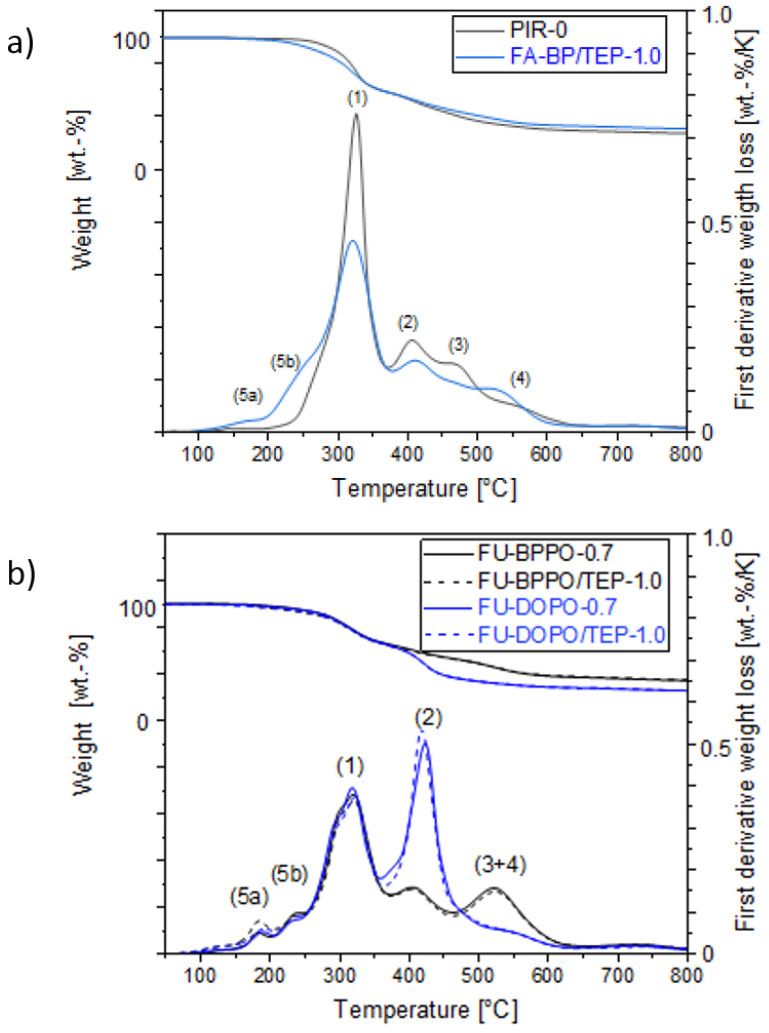
Thermogravimetric curves of selected PIR foams with phosphorus-containing FRs with different chemical structure. DOPO-containing foams can easily be identified by maximum (2) occurring in the first derivative curves (lower curves): (**a**) PIR-0 compared to a nonreactive FR and (**b**) comparison of different NCO-reactive FU-derivatives.

**Figure 4 materials-16-00172-f004:**
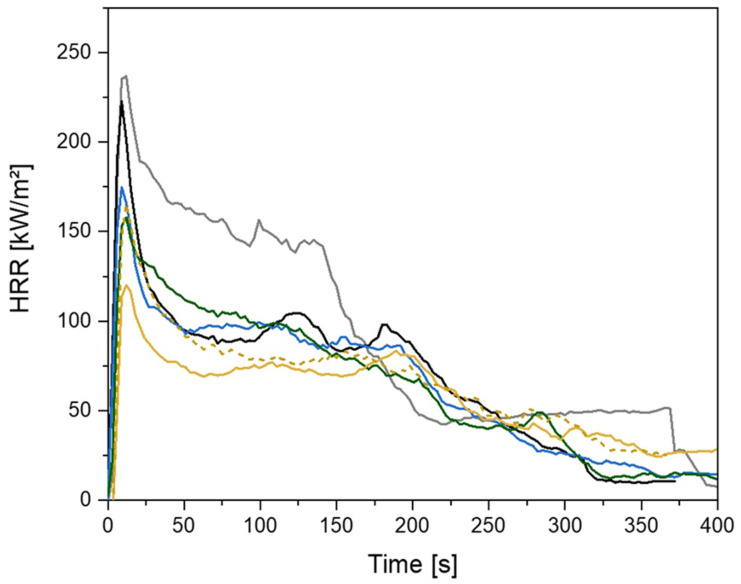
HRR vs. time curves of selected PIR foams obtained under forced flaming conditions in the cone calorimeter (50 kW·m^−2^) **(―: PIR-0; ―: TPP/TEP-1.0**; **―: EA-BPPO/TEP-1.0**; **―: FU-BPPO/TEP-1.0**; **―: TA-BPPO/TEP-1.0**; **----: TA-DOPO/TEP-1.0**).

**Figure 5 materials-16-00172-f005:**
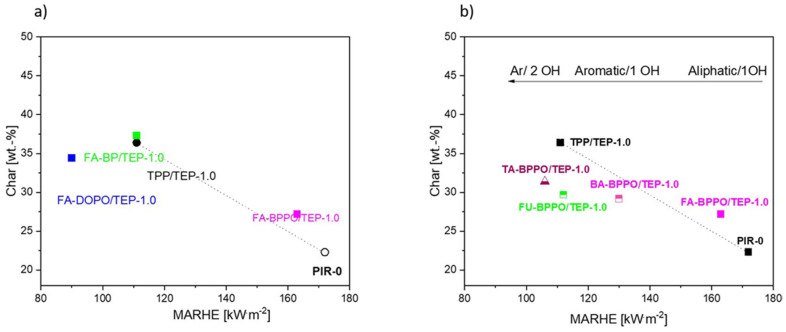
MARHE vs. char plot of (**a**) PIR foams with reactive formaldehyde (FA)-based additives and (**b**) PIR foams with reactive phospha-aldol adducts.

**Figure 6 materials-16-00172-f006:**
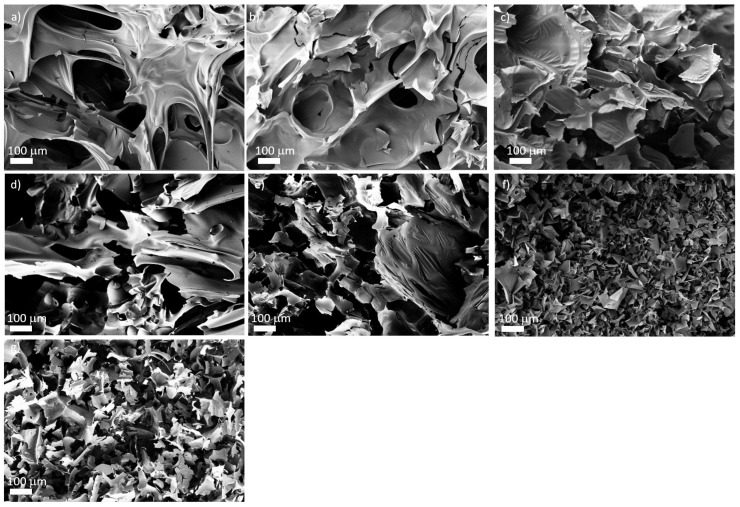
Morphology of the remaining chars after burning of PIR samples with different additives: (**a**) PIR-0; (**b**) TPP/TEP-1.0; (**c**) FA-BP/TEP-1.0; (**d**) FA-BPPO/TEP-1.0; (**e**) FA-DOPO/TEP-1.0; (**f**) BuA-DOPO/TEP-1.0; (**g**) FU-BPPO/TEP-1.0.

**Table 1 materials-16-00172-t001:** BPPO and DOPO adducts synthesized by phospha-aldol reaction and studied as FR in PUR/PIR foams: chemical structure, abbreviation, and yield.

Entry	BPPO/DOPO Added to	Chemical Structure	Abbreviation	P Content [wt.%]	Yield [%]	*M_p_* [°C]
**1a**	Paraformaldehyde	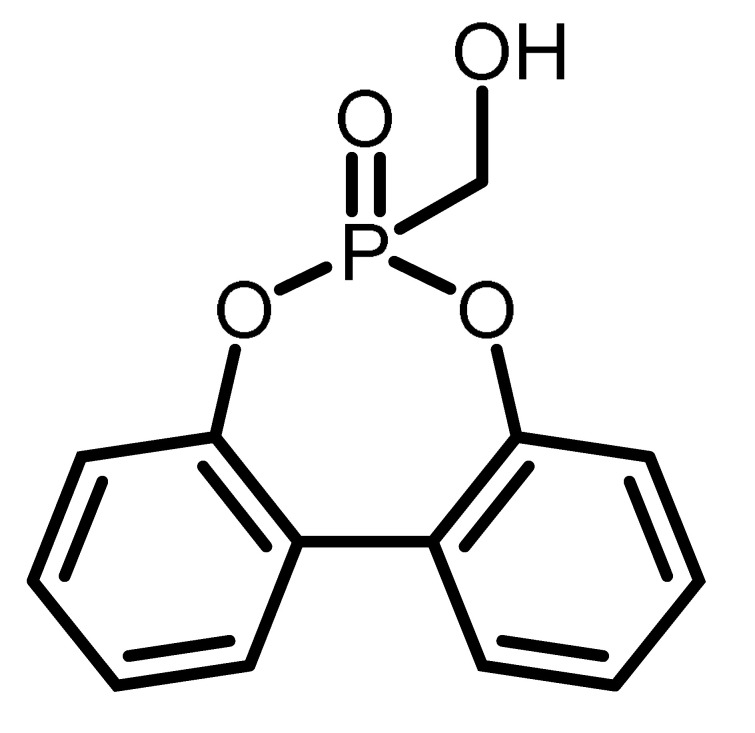	FA-BPPO	11.8	76	175
**2a**	Acetaldehyde	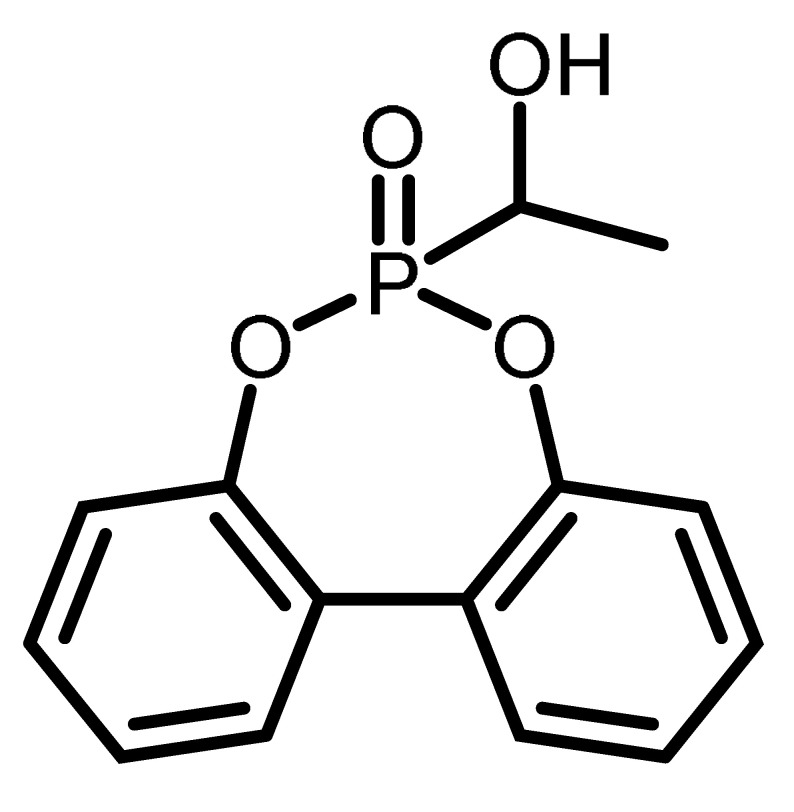	AA-BPPO	11.2	68	143
**3a**	Furfural	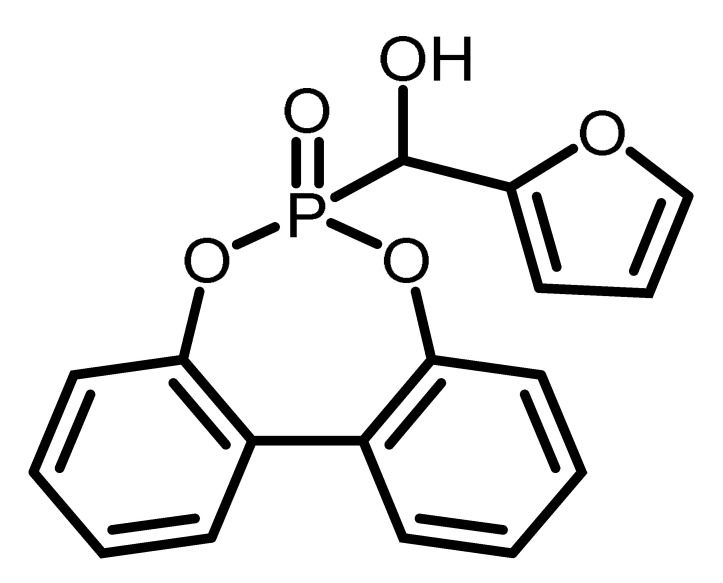	FU-BPPO	9.4	94	162
**4a**	Benzaldehyde	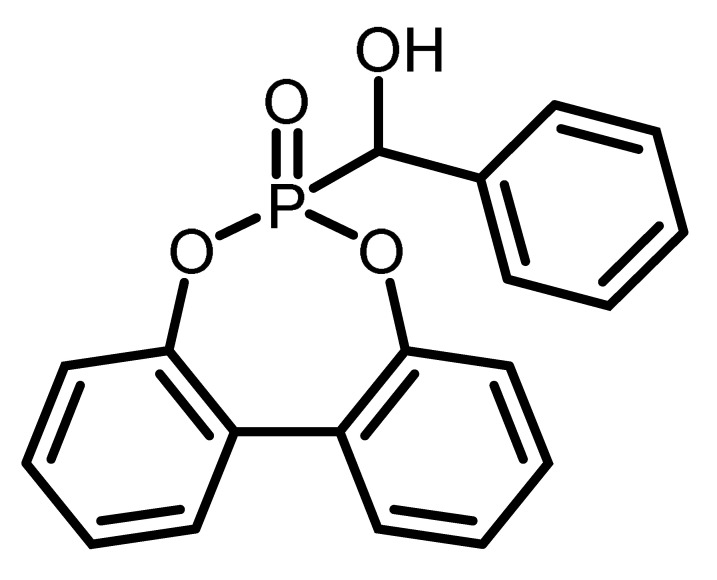	BA-BPPO	9.2	82	183
**5a**	Terephthalaldehyde	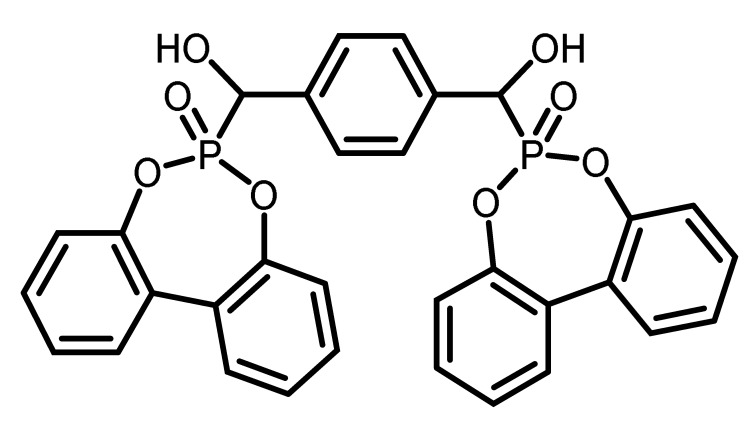	TA-BPPO	10.3	87	214
**1b**	Paraformaldehyde	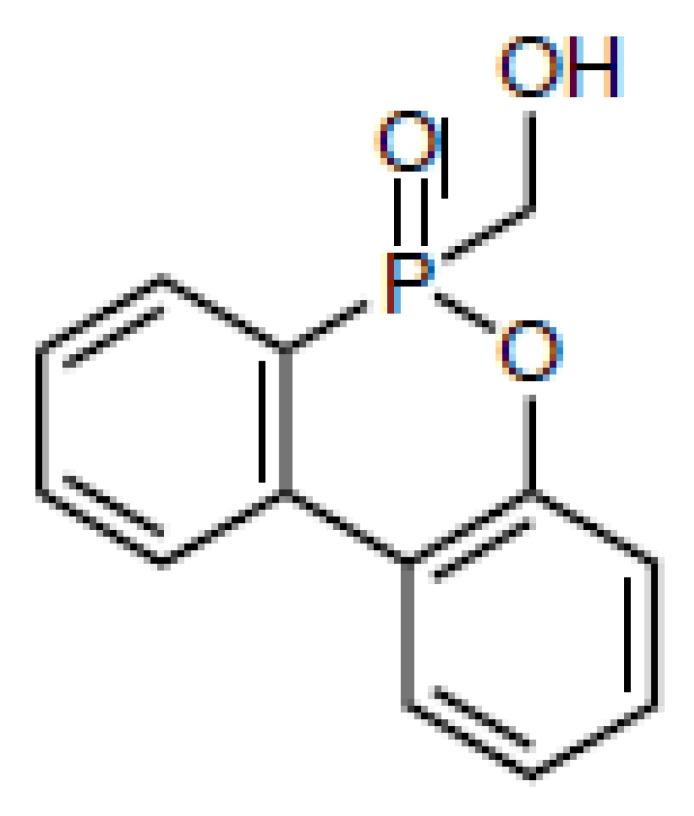	FA-DOPO	12.6	55	161
**2b**	Acetaldehyde	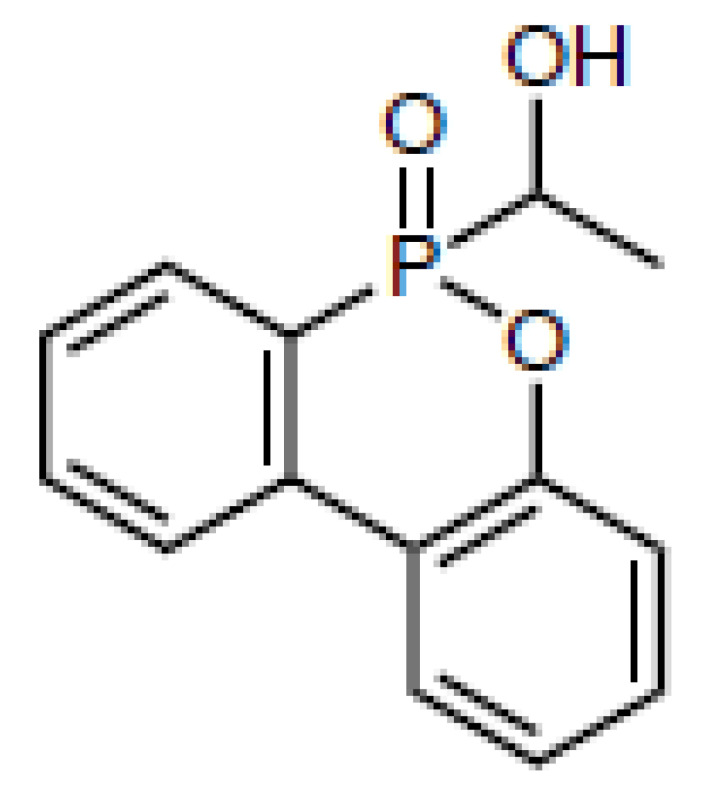	AA-DOPO	10.6	63	187
**3b**	Furfural	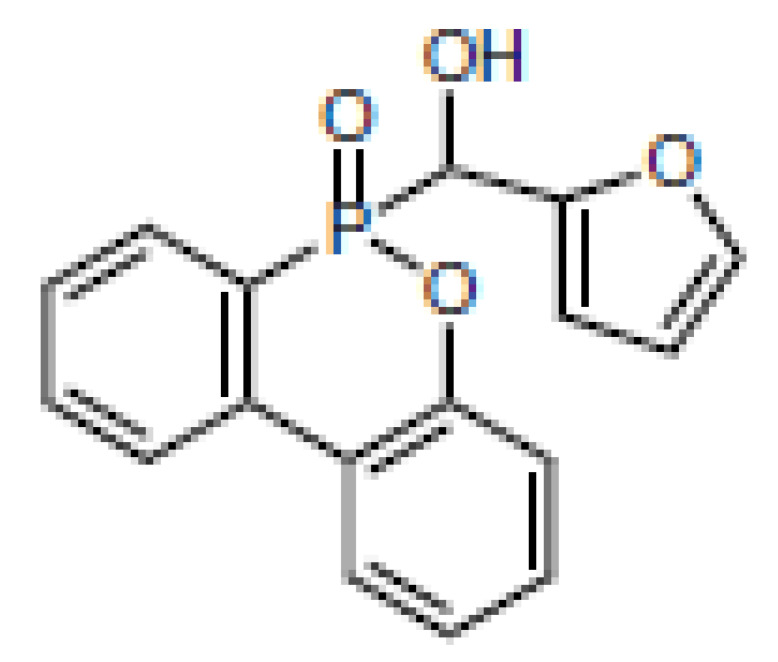	FU-DOPO	9.9	89	156
**4b**	Benzaldehyde	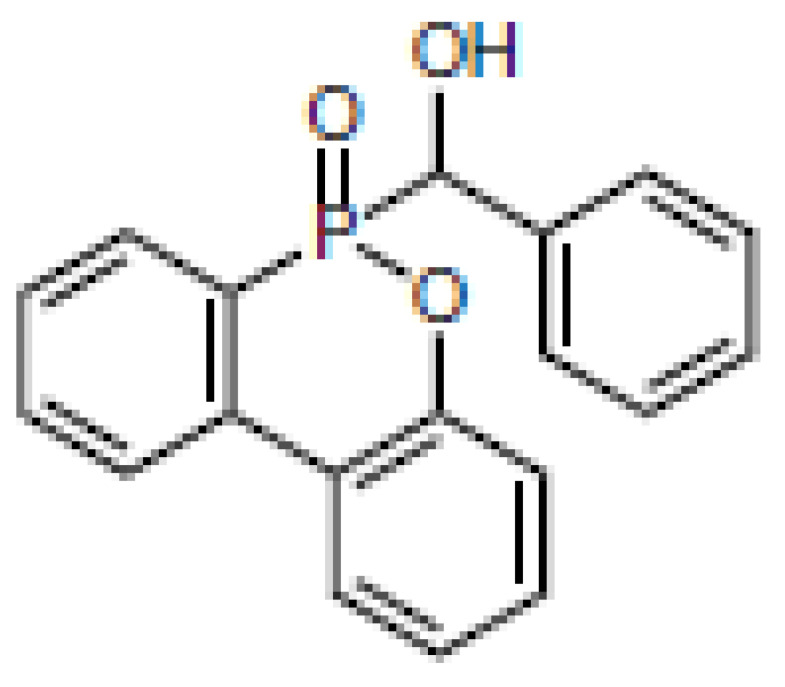	BA-DOPO	9.6	67	184
**5b**	Terepthalaldehyde	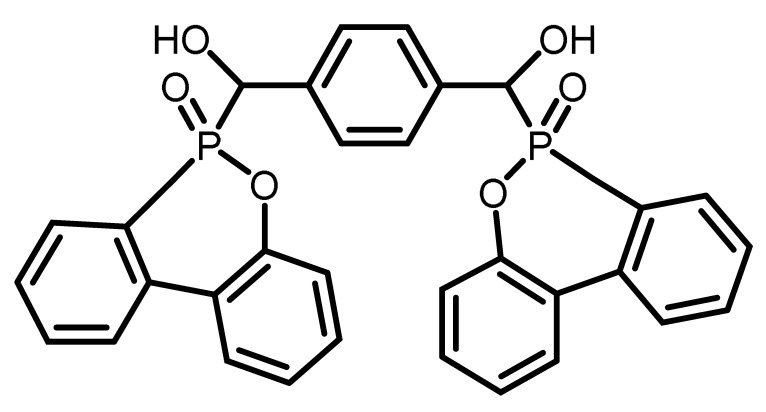	TA-DOPO	10.1	77	234

**Table 2 materials-16-00172-t002:** General formulation of the polyisocyanurate (PIR) foams studied.

Formulation Component	Amount [g]
Polyester polyol (Desmophen^®^ PEP50 AD)	53.0
Poly(ethylene glycol) PEG 400	16.0
Triethyl phosphate (TEP)	5.0
Stabilizer (TEGOSTAB^®^ B 8421)	4.0
Emulsifier (Emulsogen^®^ TS100)	2.0
Catalyst (KAc, 25 wt.% in DEG)	2.5
Blowing Agent (pentane)	15.0
Flame Retardant ^(a)^	variable
Polyisocyanate (DESMODUR^®^ 44V70L)	151.8

^(a)^ Amount depends on the P-content of the FR according to Table 1 and the P-content (1 wt.%) desired in the foam.

**Table 3 materials-16-00172-t003:** Characteristic physical properties of the PIR foams with nonreactive and reactive FRs (red: foams with densities outside the target range).

Foam Name (Contg. Used FR)	OH Groups	Comment	Density [kg⋅m^−3^]	*WA_v_* [Vol%]	Pore Size [mm]
0 (PIR-0)	0	Control	36	7	0.17
TEP-0.3	0	Control	36	4	0.18
TEP-0.7	0	Control	32	4	n. m.
TPP-0.7	0	Control	36	5	0.17
TPP/TEP-1.0	0	benchmark	38	5	0.15
EA-BPPO-0.7	0	aliphatic	36	7	0.29
EA-BPPO/TEP-1.0	0	aliphatic	37	6	0.19
PA-BPPO/TEP-1.0	0	aromatic	39	16	0.18
DMI-BPPO/TEP-1.0	0	aliphatic	35	16	0.16
DMI-DOPO/TEP-1.0	0	aliphatic	35	11	0.14
BuMA-DOPO (AC4-DOPO)-0.7	0	aliphatic	36	n. m.	n. m.
BuMA-DOPO (AC4-DOPO)/TEP-1.0	0	aliphatic	32	n. m.	n. m.
FA-BPPO-0.7	1	aliphatic	35	67	0.22
FA-BPPO/TEP-1.0	1	aliphatic	39	82	0.16
BA-BPPO-0.7	1	aromatic	47	60	0.18
BA-BPPO/TEP-1.0	1	aromatic	52	77	0.29
BA-DOPO-0.7	1	aromatic	36	91	0.15
BA-DOPO/TEP-1.0	1	aromatic	36	94	0.17
FU-BPPO-0.7	1	aromatic	41	57	0.17
FU-BPPO/TEP-1.0	1	aromatic	36	40	0.20
FU-DOPO-0.7	1	aromatic	37	98	0.14
FU-DOPO/TEP-1.0	1	aromatic	37	14	0.14
TA-BPPO-0.7	2	aromatic	35	5	0.17
TA-BPPO/TEP-1.0P	2	aromatic	41	87	0.17
TA-DOPO-0.7P	2	aromatic	36	6	0.21
TA-DOPO/TEP-1.0P	2	aromatic	36	90	0.13
TA-BP-0.7P	2	ar-aliphatic	36	76	0.17
TA-BP/TEP-1.0P	2	ar-aliphatic	36	90	0.29
FU-BP-0.7	1	ar-aliphatic	34	12	0.18
FU-BP/TEP-1.0	1	ar-aliphatic	33	19	0.21
FA-BP-0.7	1	aliphatic	34	25	0.17
FA-BP/TEP-1.0	1	ar-aliphatic	38	30	0.15
OP560-0.7	2	aliphatic	37	7	n. m.
OP560/TEP-1.0	2	aliphatic	37	n. m.	n. m.

0 OH groups: nonreactive FR; 1 OH group: reactive FR, resulting in dangling chains terminated by P-units; 2 OH groups: reactive FR, resulting (ideally) in fully incorporated P-units. n.m.: not measured; red values: outside target range.

**Table 4 materials-16-00172-t004:** Vertical flame spread data of the studied PIR foams with phosphorus-containing FRs and water uptake < 20% (PIR foams without TEP: 0.7 wt.% P; PIR foams plasticized with TEP: 1 wt.% P); flame height is given in cm.

^(a)^	NCO-Reactive Groups	DOPO	BPPO	BP	DOPO/TEP	BPPO/TEP	BP/TEP
**EA**	no	-	17	-	-	16	-
**DMI**	no	-	-	-	14	15	-
**FU**	yes	-	-	17	11	-	15
**TA**	yes	11	11	-	-	-	-

^(a)^ Combination of acrylate/aldehyde with the respective phosphorus compound; -: not examined.

**Table 5 materials-16-00172-t005:** Vertical flame spread data of the studied PIR foams with NCO-reactive phosphorus-containing FRs and water uptake > 20% (PIR foams without TEP: 0.7 wt.% P; PIR foams plasticized with TEP: 1 wt.% P); flame height is given in cm.

^(a)^	NCO-Reactive Groups	DOPO	BPPO	BP	DOPO/TEP	BPPO/TEP	BP/TEP
**FA**	1	-	18	16	-	18	14
**AA**	1	-	-	-	15	18	-
**BA**	1	12	13	-	11	12	-
**FU**	1	12	14	-	-	12	-
**TA**	2	-	-	12	14	<10	<10

^(a)^ Combination of acrylate/aldehyde with the respective phosphorus compound; -: not examined.

**Table 6 materials-16-00172-t006:** Comparison of fire-relevant parameters and structure of the chars remaining after cone calorimeter tests of closed-cell PIR control and reference foams.

Parameter	PIR-0	TEP-0.3	TPP-0.7	TEP/TPP-1.0
P content [wt.%]	0	0.3	0.7	1.0
VFS [cm]	>20	18	15	14
MARHE [kW⋅m^−2^]	172	128	145	111
Residue [wt.%]	22.3	26.7	44.7	36
Av. char height [cm]	3.0	4.0	n.m.	4.5
TSR [m^2⋅^ m^−2^]	551	392	755	324
P content in char [wt.%]	-	0.34	n.m.	0.64
P retention in char [%]	-	30	n.m.	23

n.m.: not measured.

**Table 7 materials-16-00172-t007:** Summary of fire-relevant parameters and chars remaining after cone calorimeter tests of the closed-cell PIR foams ($WA_{V}<20\%)$.

FR Foam with FR	OH gr.	Structure	Density [g⋅cm^−3^]	VFS [cm]	MARHE [kW⋅m^−2^]	TSR [m^2^ m^−2^]	CO Yield [kg⋅kg^−1^]	Residue [wt.%]	Av. Char Height [cm]	P_char_ [wt.%]	P Retention in Char ^(a)^ [%]
PIR-0	0	control	36	>20	172	551	0.04	22.3	3.0	0	-
TEP-0.3	0	control	36	18	128	392	0.05	26.7	4.0	0.34	30
TPP/TEP-1.0	0	control	39	14	111	324	0.10	36.4	4.5	0.64	23
OP560-0.7	2	al		n.m.	193	867	0.19	29.9	4.0	0.24	10
EA-BPPO-0.7	0	al	37	17	132	501	0.08	28.2		n.m.	
EA-BPPO/TEP-1.0	0	al	37	16	121	433	0.06	28.6	3.0	1.19	34
BuA-DOPO-0.7	0	al	37	n.m.	172	267	0.14	50.5	4.3	1.1	79
BuA-DOPO/TEP-1.0	0	al	37	n.m.	139	230	0.14	63.9	4.3	1.5	96
DMI-BPPO/TEP-1.0	0	al	35	15	118	317	0.04	33.2	n.m.	n.m.	
DMI-DOPO/TEP-1.0	0	a	35	14	108	371	0.05	29.4	n.m.	n.m.	
PA-BPPO/TEP-1.0	0	ar	39	11	111	360		31.3	n.m.	n.m.	
FU-BP/TEP-1.0	1	ar-al	33	15	91	298	0.28	29.5		n.m.	
FU-DOPO/TEP-1.0	1	ar	37	11	90	332	0.05	30.5	4.5	n.m.	
TA-BPPO-0.7	2	ar	35	11	86	342	0.06	31.9	3.0	n.m.	
TA-DOPO-0.7	2	ar	36	11	100	451	0.06	32.4	2.5	n.m.	

-: not available. ^(a)^ Fraction of P retained in char relative to original sample. n.m.: not measured; al: aliphatic; ar: aromatic.

**Table 8 materials-16-00172-t008:** Summary of fire-relevant parameters and chars remaining after cone calorimeter tests of the open-cell PIR foams ($WA_{V}>20\%).$

Foam with FR	OH gr.	Struct-ure	Density [g⋅cm^−3^]	VFS [cm]	MARHE [kW⋅m^−2^]	TSR [m^2^ m^−2^]	CO Yield [kg⋅kg^−1^]	Residue [wt.%]	Av. Char Height [cm]	P_Char_ [wt.%]	P Retention in Char ^(a)^ [%]
FA-BPPO-0.7	1	al	38	18	163	888	0.07	24.3	5.5	n.m.	
BA-BPPO-0.7	1	ar	47	11	173	930	0.07	28.2	3.0	0.90	36
FA-BPPO/TEP-1.0	1	al	34	18	163	754	0.06	27.2	5.5	1.04	28
FA-BP/TEP-1.0	1	ar-al	38	14	111	242	0.08	37.3	n.m.	n.m.	
BA-BPPO/TEP-1.0	1	ar	62	12	130	692	0.06	29.2	2.5	n.m.	
BA-DOPO-0.7	1	ar	36	12	108	556	0.06	29.1	2.0	n.m.	
BA-DOPO/TEP-1.0	1	ar	36	11	89	411	0.06	34.5	2.0	1.39	48
FU-BPPO-0.7	1	ar	41	14	112	455	0.08	31.6	3.5	0.97	44
FU-BPPO/TEP-1.0	1	ar	36	12	112	532	0.06	29.7	4.0	n.m.	
FU-DOPO-0.7	1	ar	37	12	105	449	0.06	31.0	3.0	0.81	36
TA-BPPO/TEP-1.0	2	ar	41	<10	106	414	0.05	31.4	n.m.	n.m.	
TA-DOPO/TEP-1.0	2	ar	36	14	106	414	0.06	31.4	4.0	n.m.	
TA-BP-0.7	2	ar-al	36	12	15	263	0.18	23.0	n.m.	n.m.	
TA-BP/TEP-1.0	2	ar-al	36	<10	14	189	0.17	26.9	n.m.	n.m.	
OP560/TEP-1.0	2	al	41	n.m.	199	823	0.20	27.0	4.2	0.65	18

-: not available. al: aliphatic; ar: aromatic; ^(a)^ Fraction of P retained in char relative to original sample; red values: outside target range.

**Table 9 materials-16-00172-t009:** Photographic images of the remaining sample residues after cone calorimetric investigations.

PIR-0 (Closed-Cell)	TEP-0.3 (Closed-Cell)	TPP-0.7 (Closed-Cell)	TEP/TPP-1.0 (Closed-Cell)
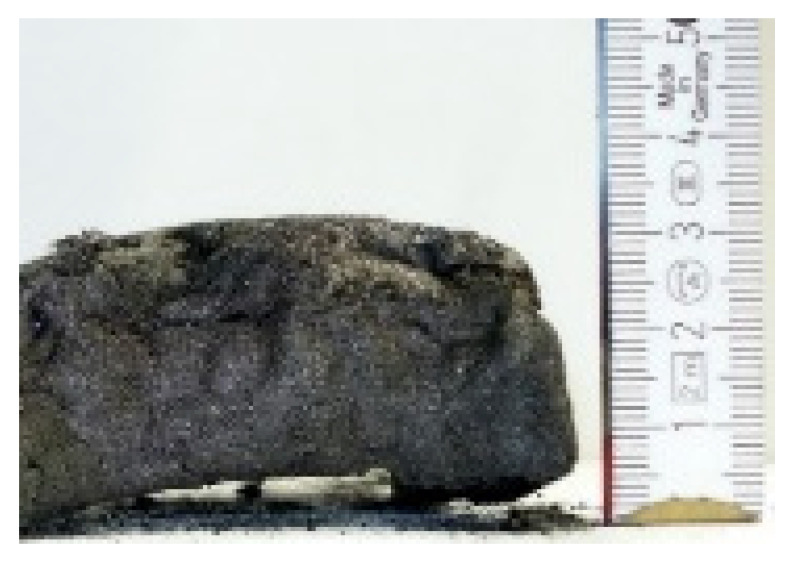	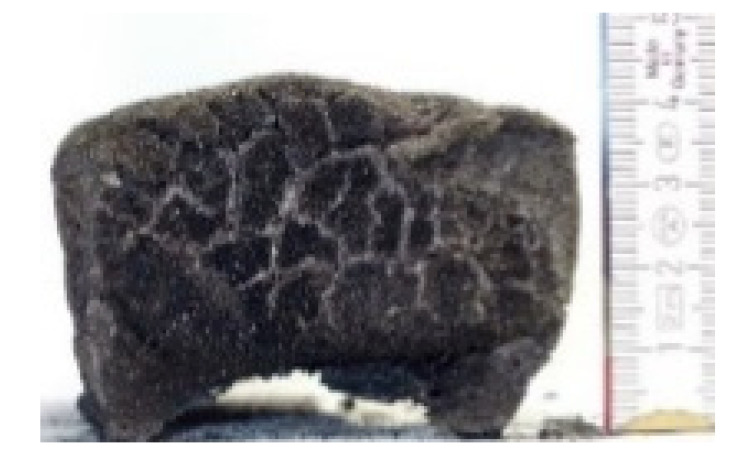	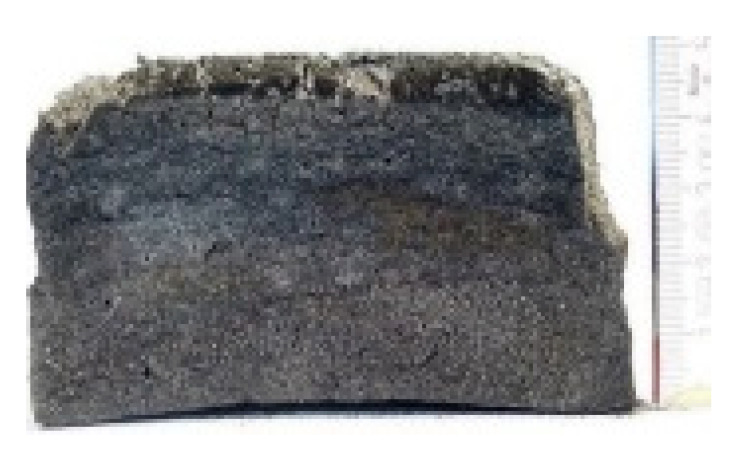	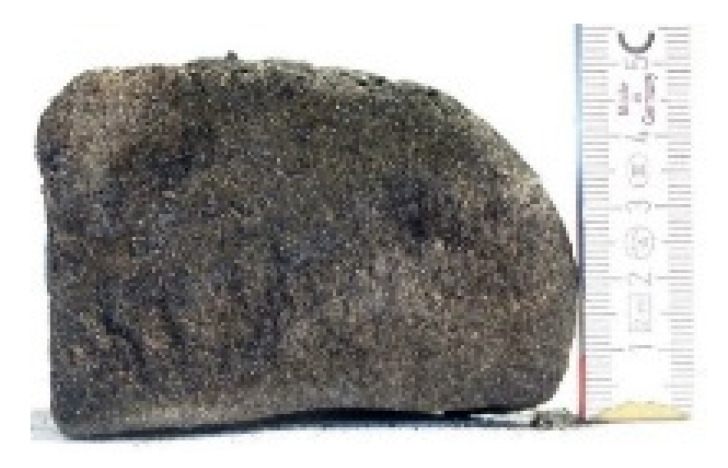
Residue: 22 wt.% P (residue): 0 wt.%	Residue: 27 wt.% P (residue): 0.34 wt.% Fraction of P retained in char to original sample: 30%	Residue: 28 wt.% P (residue): 0.24 wt.% Fraction of P retained in char to original sample: 10%	Residue: 36 wt.% P (residue): 0.64 wt.% Fraction of P retained in char to original sample: 23%
EA-BPPO/TEP-1.0 (Closed-Cell)	BuA-DOPO (AC4-DOPO)/TEP-1.0 (Closed-Cell)		
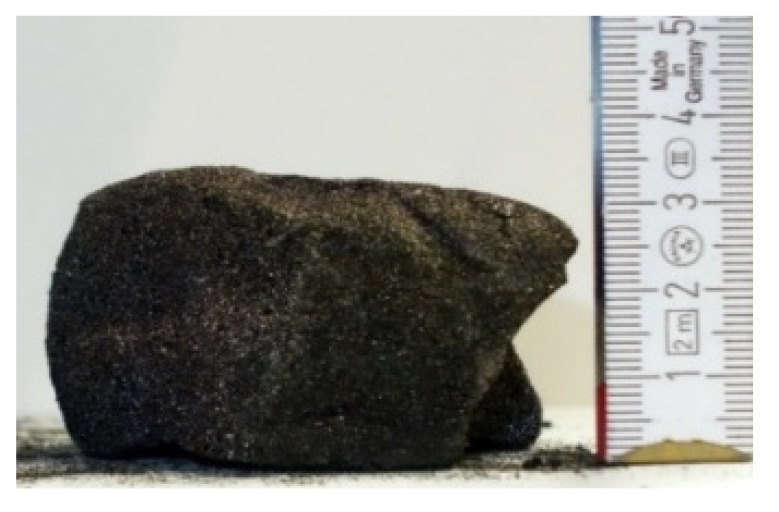	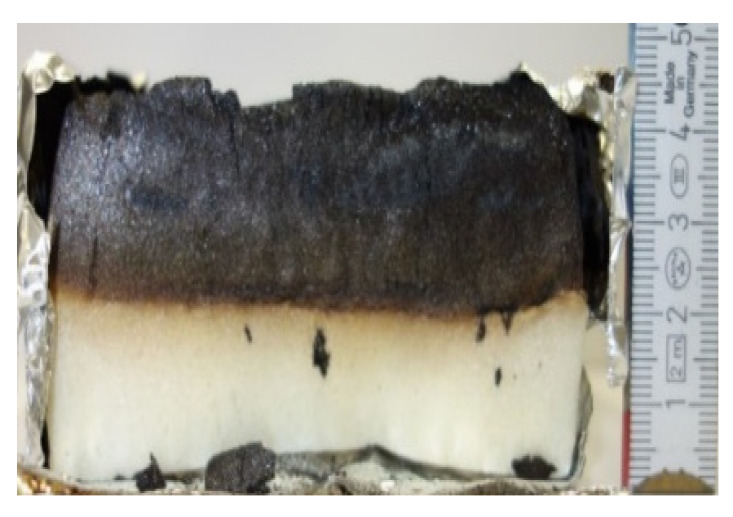		
Residue: 28.6 wt.% P (residue): 1.19 wt.% Fraction of P retained in char to original sample: 34%	Residue: 63.9 wt.% P (residue): 1.50 wt.% Fraction of P retained in char to original sample: 60%		
FA-BP/TEP-1.0 (Open-Cell)	FA-BPPO/TEP-1.0 (Open-Cell)	FA-DOPO/TEP-1.0	
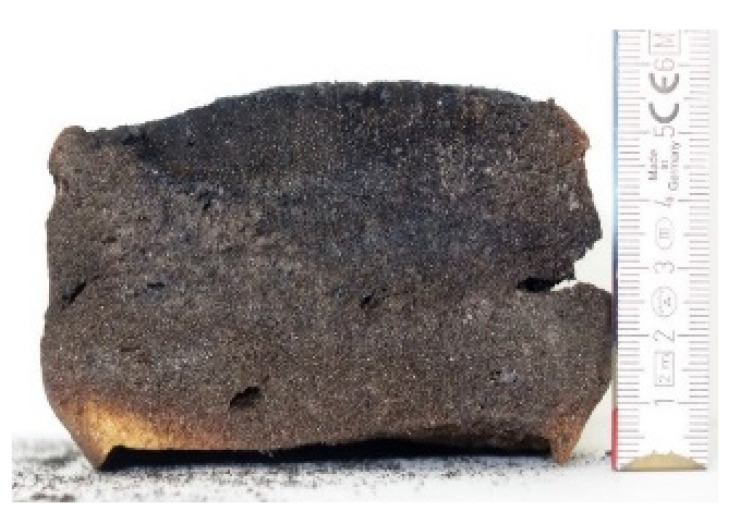	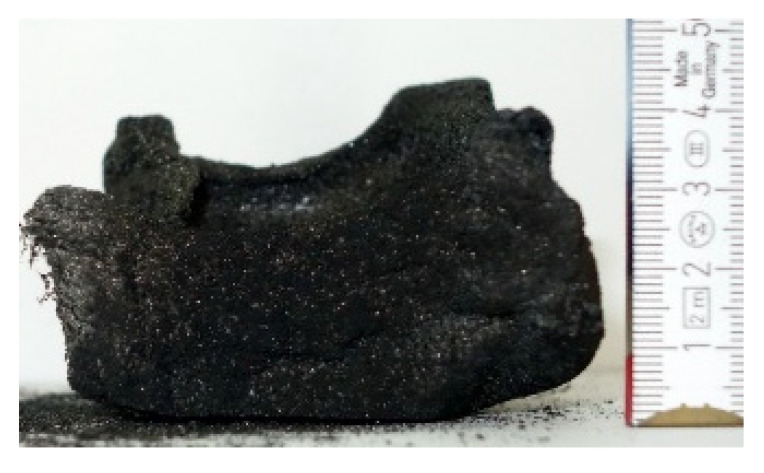	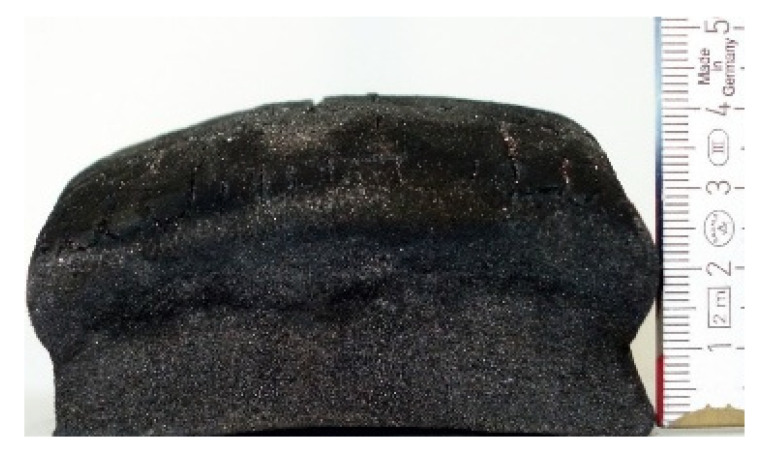	
Residue: 37.5 wt.%	Residue: 27.2 wt.% P (residue): 1.04 wt.% Fraction of P retained in char to original sample: 28%	Residue: 34.4 wt.%	
FU-BPPO/TEP-1.0 (Open-Cell)	FU-DOPO/TEP-1.0 (Closed-Cell)	TA-BPPO/TEP-1 (Open-Cell)	TA-DOPO/TEP-1.0 (Open-Cell)
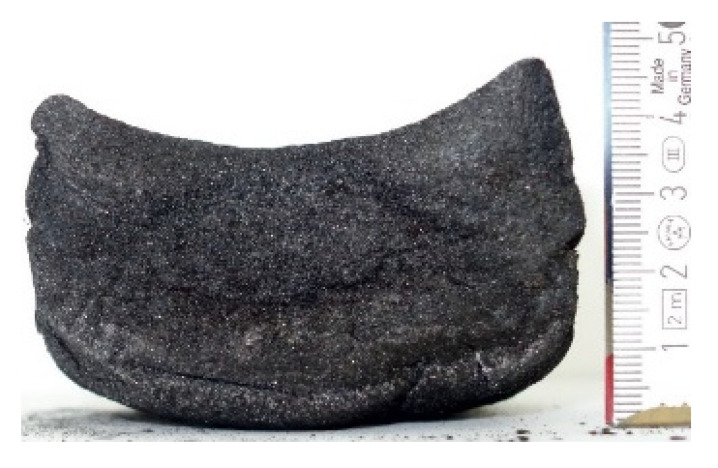	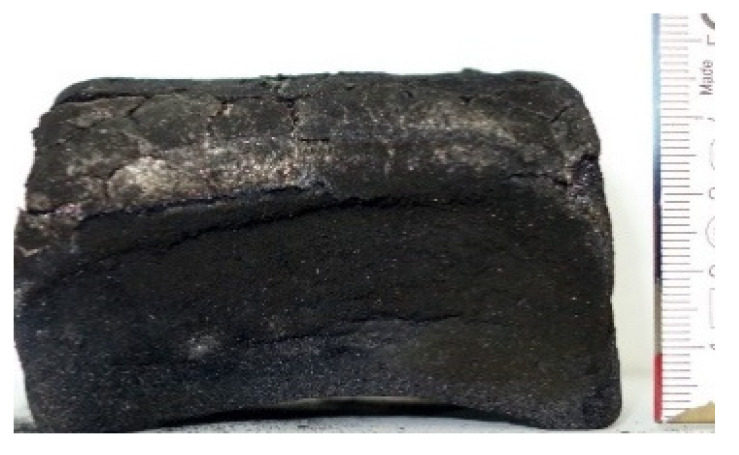	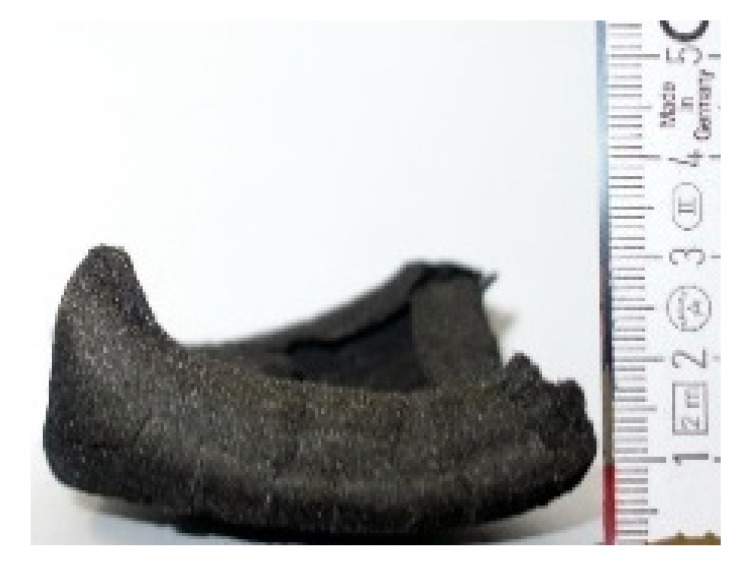	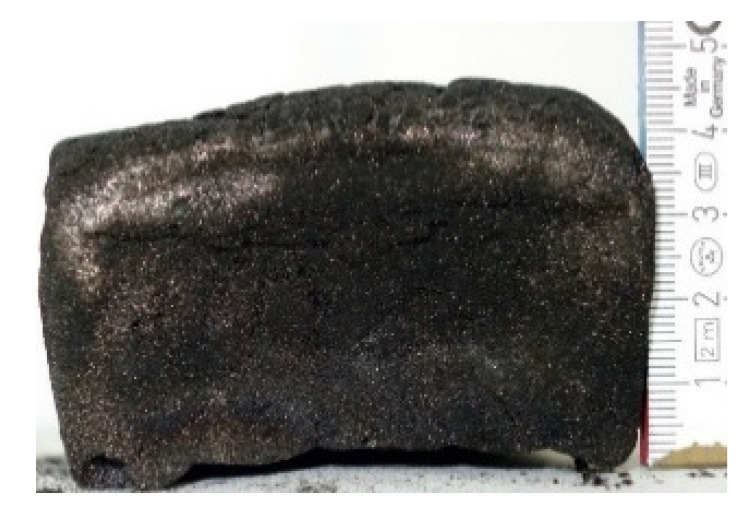
Residue: 29.7 wt.%	Residue: 30.5 wt.%	Residue: 31.4 wt.%	Residue: 31.4 wt.%

**Table 10 materials-16-00172-t010:** Schematic representation of the P-containing FRs incorporated in different manners into rigid PIR foams, typical examples studied, and summarized result.

A	B1	B2	C
Not chemically bound but dissolved in the polymer matrix	Chemically bound to the polymer matrix as chain terminator	Dangling to the polymer matrix as substituent	Part of the polymer chains
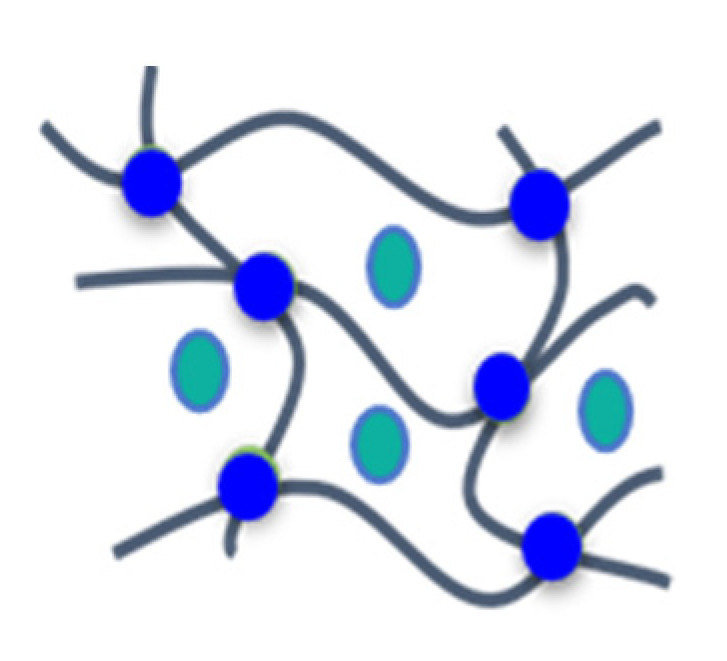	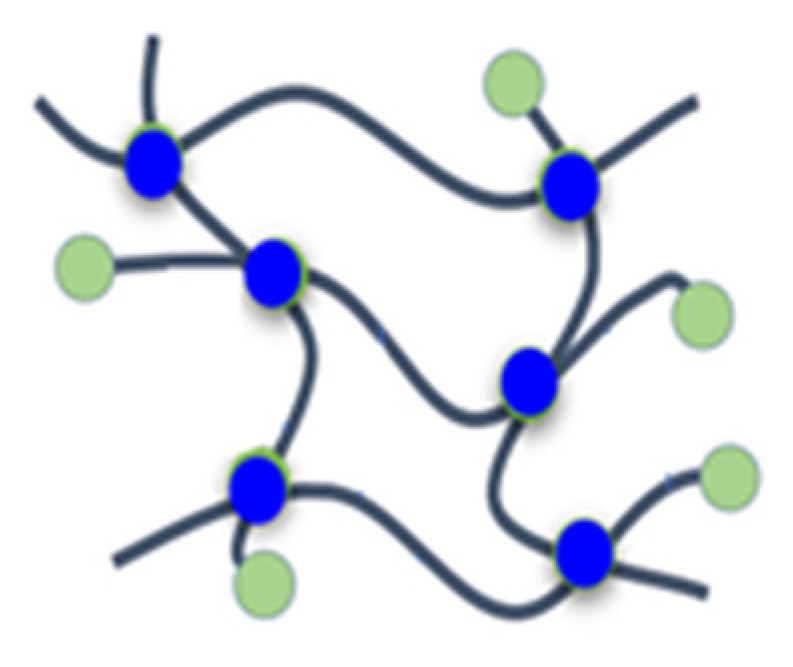	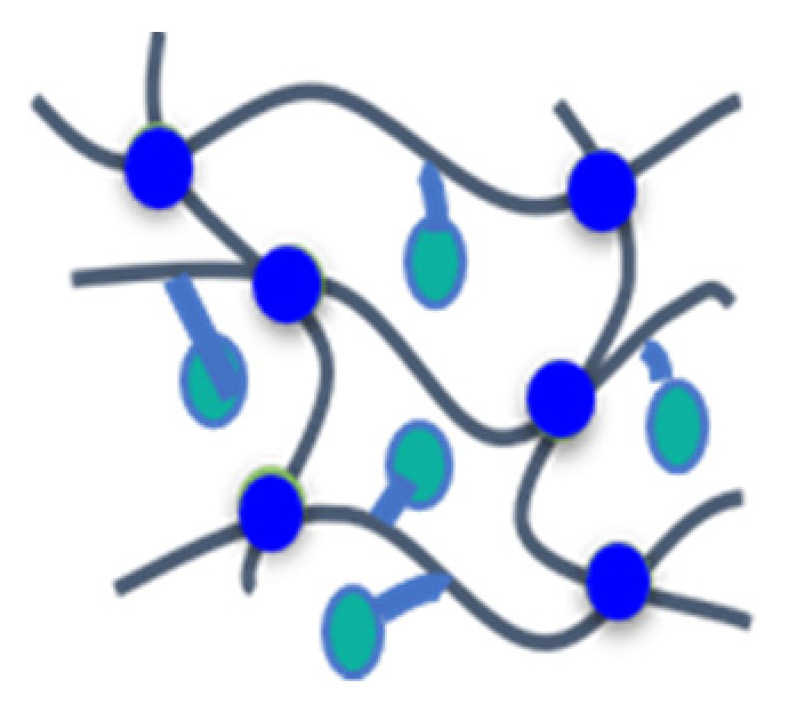	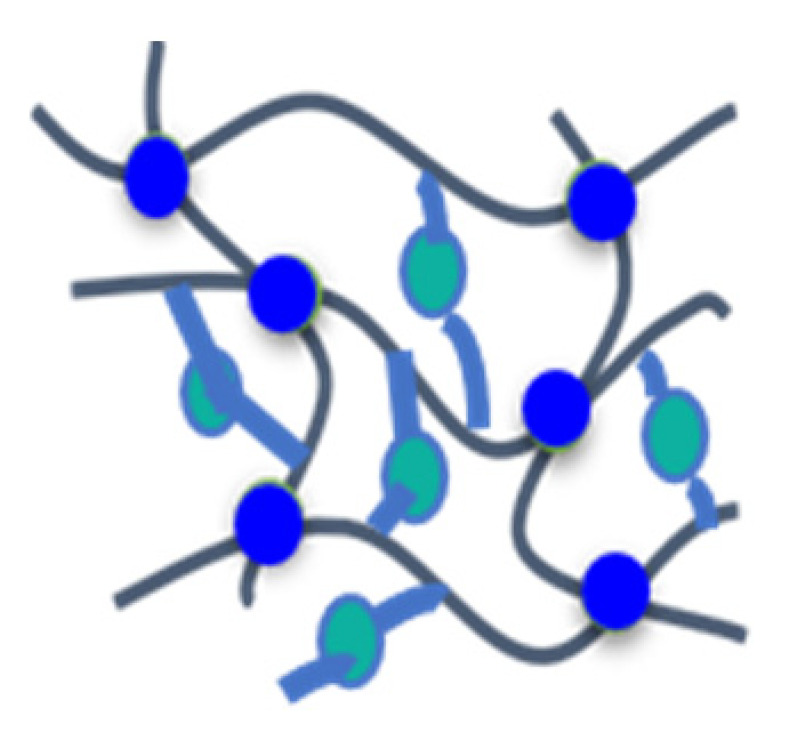
TEP, TPP, EA-BPPO, DMI-BPPO, BuA-DOPO	FU-BPPO, FU-DOPO, AA-BPPO	TA-DOPO, TA-BPPO	Exolit^®^ OP 560
Closed-cell morphology, moderate effect on FR	Significant positive effect on FR	Significant positive effect on FR	No or weak effect on FR

## Data Availability

The detailed data of this study are summarized in the Appendix A.

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
