# Peer review of "Effective Halogen-Free Flame-Retardant Additives for Crosslinked Rigid Polyisocyanurate Foams: Comparison of Chemical Structures"

_materials, 2022, doi:10.3390/ma16010172_

Round 1
Reviewer 1 Report
There are too more contents and the compression is necessary.
(1) It is better that Figure 5, Figure 6 and Figure 7 are deleted
(2) The schematic structures of the resulting foams summarized and illustrated in Table 9 should be inserted in part 3. Results and discussion.
(3) These endnotes such as: EA: ethyl acrylate; PA: phenyl acrylate;…… in Table 3 and Table 7 should be deleted.
Author Response
First of all, we would like to express our thanks to the editors and reviewers for taking their time to evaluate the manuscript. We took all comments carefully into account and revised the manuscript accordingly. The revisions were done in track mode as requested. The replies to the reviewer comments given in the following are marked red (also enclosed as attachment)
Reviewer 1
Open Review
There are too more contents and the compression is necessary.
- It is better that Figure 5, Figure 6 and Figure 7 are deleted
Authors comment:
We agree with the reviewer that the manuscript is very big. This is due to the extensive study done and the high number of interesting results. However, we followed the suggestion of the reviewer and deleted Figure 5 and Figure 6. These figures were shifted into the Supplementary Information (SI) as Figures SI-28 and SI-29 in the new section 6. Cone calorimeter results: MARHE vs. char plots. Figure 7 was kept in the main part as Figure 5 and all figures were renumbered. The jpeg files of the figures were also renumbered and will be submitted again together with the revised version.
- The schematic structures of the resulting foams summarized and illustrated in Table 9 should be inserted in part 3. Results and discussion.
Authors comment:
Checking this suggestion it was found that Table 9 is actually Table 10 – this was corrected. This table was intended to give the reader a schematic summary of the different systems investigated and the respective results obtained. From that point of view, it clearly belongs to the Conclusion paragraph. The respective sentence was extended to “The schematic structures of the resulting foams and their impact on the flame and fire retardant properties are summarized and illustrated in Table 10.”
The Results and Discussion section ends with a discussion of the smoke release. Another subparagraph would have to be added to place Table 10 in that section. Therefore, it appears better to keep it in the Conclusion section.
(3) These endnotes such as: EA: ethyl acrylate; PA: phenyl acrylate;…… in Table 3 and Table 7 should be deleted.
Authors comment:
Thank you for the suggestion. We deleted the abbreviations (footnotes) under Table 3, Table 7 and Table 8. The abbreviations were introduced in section 2.2.1. (phospha-Michael adducts) and were already part of Table 1 (phospha-aldol adducts).
Thank you again for your time and we hope that the manuscript is now ready for publication.
Kind regards,
Doris Pospiech

Reviewer 2 Report
1. Throughout the manuscript there are messages: Error! Reference source not found. Please fix it.
2. Page 11 Last sentence. Please change open-cell to closed-cell
3. Figure 2 Title, at scale bar 200 µm, there is a symbol before m instead of µ. Also, it would be easier for readers to add open or closed at each picture or just O and C.
4. Page 14. Authors wrote:” The reactive phospha-aldol FRs based on ali[1]phatic aldehydes resulted in foams with unsatisfactory VFS. If aromatic aldehydes were used, the resulting FR imparted very good flame retardant properties.” Please add in the text what is considered as unsatisfactory VFS and what as satisfactory or very good.
5. Page 14. Authors wrote: ”This supports our previous conclusion that crosslink density and FR performance tend to correlate.” Please add short discussion on influence of the chemical structure of the aldehyde (aromatic, furfural, aliphatic) in BPPO adducts on VFS.
Author Response
Reviewer 2
First of all, we would like to express our thanks to the editors and reviewers for taking their time to evaluate the manuscript. We took all comments carefully into account and revised the manuscript accordingly. The revisions were done in track mode as requested. The reply to the reviewer comments given in the following are marked red and are contained complete in the attachment.
(x) English language and style are fine/minor spell check required
Author comment: The spell check was carefully carried out throughout the revised version.
- Throughout the manuscript there are messages: Error! Reference source not found. Please fix it.
Author comment:
The manuscript version submitted by the editors of Materials did not contain these error messages. We checked carefully again and did not find any of them. These error messages were found in the pdf created by us, and we would like to apologize for that!
- Page 11 Last sentence. Please change open-cell to closed-cell
Author comment:
Thank you very much for the advice. It is of course completely right and it was changed.
- Figure 2 Title, at scale bar 200 µm, there is a symbol before m instead of µ. Also, it would be easier for readers to add open or closed at each picture or just O and C.
Author comment:
Thank you very much, the wrong symbol was deleted and substituted by “micro”. With respect to the second point we would like to state that we prefer to maintain the figure as it is. All changes in the file often reduce the quality. The reader should be able to bring these two infos (image and caption) together. We hope that this is ok for publication.
- Page 14. Authors wrote:” The reactive phospha-aldol FRs based on ali[1]phatic aldehydes resulted in foams with unsatisfactory VFS. If aromatic aldehydes were used, the resulting FR imparted very good flame retardant properties.” Please add in the text what is considered as unsatisfactory VFS and what as satisfactory or very good.
Author comment:
You are right, this information was missing. We added the following sentences at the beginning of section 3.4.1.:
“The VFS test according to DIN 4102 can be considered as first measure for evaluation of the burning behavior. The flame height given in cm decides about the classification into the categories “B2 classification passed” (satisfactory properties, flame height ≤ 15 cm) or “not passed” (flame heights > 15 cm).”
- Page 14. Authors wrote: ”This supports our previous conclusion that crosslink density and FR performance tend to correlate.” Please add short discussion on influence of the chemical structure of the aldehyde (aromatic, furfural, aliphatic) in BPPO adducts on VFS.
Author comment:
Thank you. This discussion was already given in section 3.4.1. However, we tried to reformulate the text to make the results very clear for the reader (see manuscript in track mode).
Taking all the suggestions of the reviewers into account and carefully revising the manuscript we hope that it is now ready for publication in Materials.
Kind regards,
Doris Pospiech

Round 2
Reviewer 1 Report
The paper presented the results of systematic studies for non-reactive and reactive phosphorus-containing FRs in PIR foams. It is found that the subtle variation in the chemistry around the P atom caused significant changes in the FR effect.The investigations can provide beneficial and valuable help for the choice of FRs in PIR foams. It can be accepted to publish in Materials.